# Quantitative metaproteomics of medieval dental calculus reveals individual oral health status

Rosa R. Jersie-Christensen[1], Liam T. Lanigan[2], David Lyon[3], Meaghan Mackie [1,2], Daniel Belstrøm[4], Christian D. Kelstrup [1], Anna K. Fotakis [2], Eske Willerslev[5,6], Niels Lynnerup[7], Lars J. Jensen[3], Enrico Cappellini [2] & Jesper V. Olsen [1]

The composition of ancient oral microbiomes has recently become accessible owing to advanced biomolecular methods such as metagenomics and metaproteomics, but the utility of metaproteomics for such analyses is less explored. Here, we use quantitative metaproteomics to characterize the dental calculus associated with the remains of 21 humans retrieved during the archeological excavation of the medieval (ca. 1100–1450 CE) cemetery of Tjærby, Denmark. We identify 3671 protein groups, covering 220 bacterial species and 81 genera across all medieval samples. The metaproteome profiles of bacterial and human proteins suggest two distinct groups of archeological remains corresponding to health-predisposed and oral disease-susceptible individuals, which is supported by comparison to the calculus metaproteomes of healthy living individuals. Notably, the groupings identified by metaproteomics are not apparent from the bioarchaeological analysis, illustrating that quantitative metaproteomics has the potential to provide additional levels of molecular information about the oral health status of individuals from archeological contexts.

[1] Proteomics Program, Novo Nordisk Foundation Center for Protein Research, Faculty of Health and Medical Sciences, University of Copenhagen, Blegdamsvej 3B, 2200 Copenhagen N, Denmark. [2] Natural History Museum of Denmark, University of Copenhagen, Øster Voldgade 5-7, 1350 Copenhagen K, Denmark. [3] Disease Systems Biology Program, Novo Nordisk Foundation Center for Protein Research, Faculty of Health and Medical Sciences, University of Copenhagen, Blegdamsvej 3B, 2200 Copenhagen N, Denmark. [4] Periodontology and Microbiology, Department of Odontology, Faculty of Health Sciences, University of Copenhagen, Nørre Allé 20, 2200 Copenhagen N, Denmark. [5] Centre for GeoGenetics, Natural History Museum of Denmark, University of Copenhagen, Øster Voldgade 5-7, 1350 Copenhagen K, Denmark. [6] Department of Zoology, University of Cambridge, Downing St, Cambridge CB2 3EJ, UK. [7] Laboratory of Biological Anthropology, Institute of Forensic Medicine, Faculty of Health Sciences, University of Copenhagen, Frederik V's Vej 11, 2100 Copenhagen Ø, Denmark. These authors contributed equally: Rosa R. Jersie-Christensen, Liam T. Lanigan. These authors jointly supervised this work: Enrico Cappellini, Jesper V. Olsen. Correspondence and requests for materials should be addressed to E.C. (email: ecappellini@snm.ku.dk) or to J.V.O. (email: jesper.olsen@cpr.ku.dk)

Recent investigations have shown that dental calculus (mineralized plaque) from archeological samples is a rich source of biomolecules[1–4]. Calculus preserves ancient biomolecules that relate to diet[5], and preserve a lifelong reservoir of the oral microbiome, as well as proteins and DNA from the host[6,7]. The human oral microbiome is a complex system that consists of ~700 bacterial species, as well as fungi, viruses, and archaea[8]. Furthermore, it is highly individual, and the influence of lifestyle, hygiene, environment, genetics, diet, and disease on bacterial composition has yet to be well understood[9–11].

Different molecular techniques have been used to characterize the human oral microbiome, but the vast majority of studies are based on 16S ribosomal RNA sequencing[12,13]. Using mass spectrometry (MS)-based proteomics profiling as a tool for characterizing the oral microbiome is still unconventional. However, it has the potential to provide information about the functional and active microbiome through revealing the levels of individual proteins expressed by different organisms. Furthermore, metaproteomics can identify the proteins expressed by the host, and thus, elucidate possible interactions between the host and potential pathogenic species.

Based on an optimized sample preparation protocol for metaproteomics of ancient dental calculus and state-of-the-art mass spectrometry technology, we here analyze 22 dental calculus samples from 21 archeological individuals. To identify potential divergence from the medieval microbiome, modern samples of calculus and plaque are collected from seven healthy volunteers and analyzed with identical methodology. The overall aim of the presented study is to characterize the Danish medieval oral microbiome by proteomics, in order to learn more about the individualized oral health, and possibly diet, in this specific population, as well as to compare the results to modern healthy individuals.

We observe that the set of individuals we investigated can be divided in two groups: one health-predisposed and another more susceptible to oral disease. In both groups, the oral microbiome is more heterogeneous than in modern Danish individuals. We also use high pH reversed-phase fractionation in combination with TMT labelling. We show it greatly improves sensitivity for identifying more peptides in archeological samples, highlighting the potential of this strategy for future quantitative proteomics analyses of archeological remains.

## Results

**Description of the archeological site.** The 22 samples come from a medieval parish cemetery located in Tjærby, Jutland, Denmark (Fig. 1). While there is evidence of a wooden church from around 1050 CE, all burials used in this study date from the establishment of a Romanesque stone church approximately a century later until its abandonment during the Reformation (ca. 1537 CE)[14]. The site represents an ordinary Danish medieval village, and thus the individuals should show a relative degree of uniformity in terms of lifeways and social status, allowing the recovery of proteomes that are fairly comparable.

This site was chosen because the material is well-curated and easily accessible, but also of interest because it is one of few Danish cemeteries from the medieval period that has been fully excavated. Tjærby, in its mundaneness, gives insight into the Danish medieval oral microbiomes of a relatively large number of average individuals. By studying individual oral microbiomes, future comparison with other assemblages, either contemporary or more ancient, should be more nuanced.

In order to look at the diversity in microbiome, as well as disease biomarkers from dental calculus, we were interested in individuals most likely to have osseous changes and destruction

of the alveolar bone related to periodontitis. Older males were selected based on the understanding that males often have a more aggressive inflammatory immune response compared to females[15–17]. Males over 45 with a maxilla, mandible, teeth, evidence of periodontal disease, and sufficient dental calculus for

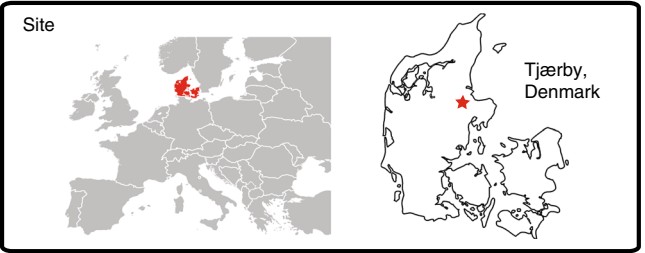

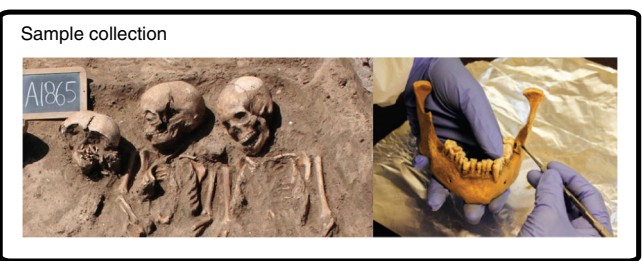

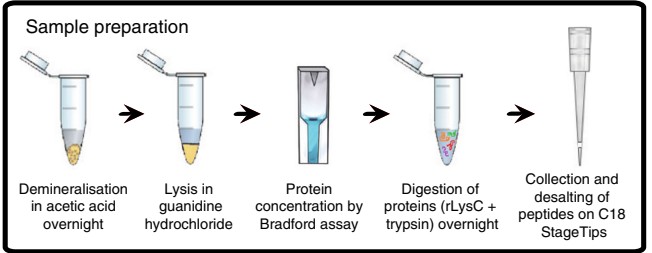

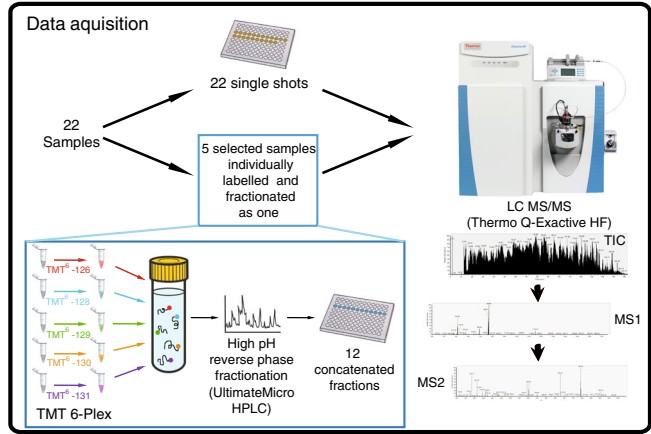

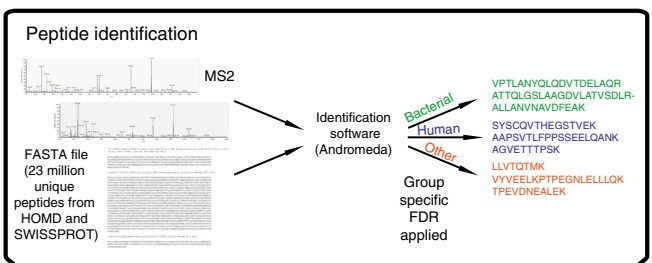

**Fig. 1** Analysis workflow. Shown are the site location of medieval samples, sample preparation strategy and data analysis pipeline. Credits: excavation: Østjylland Museum (see acknowledgements), mandible: AKF

sampling, were the initial criteria for inclusion. However, this yielded only 12 samples. By adding younger males, it was possible to collect 22 dental calculus samples from 21 individuals (further information can be found in Supplementary Note 1 and 2, and Supplementary Fig. 1). All individuals displayed some degree of periodontitis, a polymicrobial infection that leads to a sustained inflammatory response by the host's immune system. Periodontitis is complicated in its presentation, as it can occur either as a continuous or episodic breakdown of the supporting structures of the teeth, and may be locally aggressive [18]. The damage to the periodontium is caused by toxins and waste products of biofilm bacteria and dysregulation of the host's immune response, including overexpression of cytokines, prostaglandins, and other cell-derived immune-response mediators[19].

**Bioarchaeological analysis.** The 21 skeletons used in this study were generally well preserved, as documented by the full osteological analysis performed for each individual. Age-related conditions like degenerative joint disorder, osteoarthritis, and rotator cuff disease were present, which is to be expected in older individuals. The Bradford classification system[20] was used to assess the periodontal health status of the individuals. This involves examination of the buccal (cheek side) alveolar bone and the use of a periodontal probe to identify any osseous (bony) troughs between the alveolus and the tooth (Supplementary Fig. 2). Bradford scores between 1 and 4 were assigned per tooth according to severity of apparent disease state. In order to select truly periodontic individuals, those with Bradford scores of 2 were dismissed as being within the range of health, as some pathogenic alveolar bone crest changes are to be expected in older medieval individuals naïve to dental care. Therefore, Bradford scores of 3, where an infrabony trough of 2–4 mm was present were considered the baseline for more advanced periodontitis, and Bradford scores of 4, where a trough of >5 mm depth was present, were considered severe. Once a periodontal pocket of 5 mm or greater forms, it is considered a clinical risk-assessment sign of periodontitis.

Individuals were considered the least periodontally healthy when more than half of the observations were assigned scores of 3, or any scores of 4 were present. Similarly, scores were given to individuals that had one or more gross caries (Supplementary Fig. 3), and when more than two teeth were lost antemortem. Widespread periodontium involvement, deep pocket formation,

and gross carious lesions are all risk factors to the ultimate adverse outcome, the loss of the tooth or its functionality. The different scores and observations were evaluated and combined into a single score. Minor caries, limited antemortem tooth loss (AMTL), and some slight periodontal pocket formation, can be considered within the range of normal medieval health. The resulting Pathology Score (Supplementary Table 1) varies from one individual with three positive scores (unhealthiest) to five individuals with three negative scores, which can be considered the healthiest for this assemblage.

**Metaproteomics workflow.** The MS-based proteomics workflow used in this study resembles in many ways a typical workflow used for analysis of modern samples, with some exceptions (Fig. 1). Briefly, the collected dental calculus was demineralized in acetic acid, and protein extraction was subsequently performed by boiling in a GndHCl buffer. Proteins were digested in-solution by endoproteinase Lys-C and trypsin overnight. The resulting peptide mixtures were analyzed by single-shot nanoflow liquid chromatography tandem mass spectrometry (nanoLC–MS/MS) on a Q-Exactive HF orbitrap mass spectrometer with optimized fill times for parallel acquisition[21]. All acquired raw LC–MS/MS files were processed using the MaxQuant software suite[22]. Peptides were identified by searching tandem mass spectra against a combined database of the Human Oral Microbiome Database[23] and the complete SwissProt database[24]. Peptides were separated into three taxonomic groups: bacterial, human, and other. The last category comprised of food-related proteins and proteins from non-bacterial prokaryotes (Archaea). False discovery rate (FDR) calculations were performed individually within each of these groups achieving an estimated FDR of one percent. Label-free protein quantitation (LFQ) based on the MaxLFQ algorithm was employed to perform comparative analyses between all samples. To compare the metaproteome of medieval samples with that of modern material, we also analyzed proteins from plaque and calculus from modern healthy volunteers using the same protocol and analytical workflow. The results are described based on the three aforementioned taxonomic groups.

**Overview of the metaproteomics results.** When excluding proteins found only in modern samples, we identified a total of 3671 protein groups based on at least two peptides (unique + razor) (Table 1). Between 85 and 95% of the identified proteins across

**Table 1 Number of identified proteins and peptides in each sample type and distribution among the three protein categories**

|  | Bacteria | Human | Other | Total |
|---|---|---|---|---|
| Tjærby (n = 22) |  |  |  |  |
| PSM |  |  |  | 221,063 |
| Peptides, total | 28,265 | 2685 | 107 | 31,044 |
| Peptides, median (std) | 4343 (±2147) | 611 (±184) | 23 (±11) | 4930 (±2281) |
| Proteins, total | 3454 | 205 | 12 | 3671 |
| Proteins, median (std) | 781 (±392) | 69 (±18) | 4 (±1) | 854 (±404) |
| Modern plaque (n = 7) |  |  |  |  |
| PSM |  |  |  | 106,830 |
| Peptides, total | 18,607 | 7940 | 1880 | 28,023 |
| Peptides, median (std) | 5494 (±2146) | 3869 (±981) | 784 (±255) | 9846 (±2554) |
| Proteins, total | 2510 | 496 | 180 | 3186 |
| Proteins, median (std) | 952 (±316) | 312 (±63) | 111 (±29) | 1432 (±336) |
| Modern calculus (n = 6) |  |  |  |  |
| PSM |  |  |  | 153,017 |
| Peptides, total | 36,108 | 9273 | 2317 | 47,422 |
| Peptides, median (std) | 10,510 (±2392) | 2987 (±1829) | 609 (±503) | 14,928 (±1493) |
| Proteins, total | 3665 | 493 | 210 | 4368 |
| Proteins, median (std) | 1335 (±357) | 298 (±90) | 87 (±39) | 1834 (±322) |

the medieval samples were of bacterial origin and 4–14% of human origin, while less than one percent could be assigned to other taxa. The observed fractional percentage of human proteins is in the same range as the 15% reported in a previous study of archeological dental calculus[6], as well as in accordance with investigations of modern plaque biofilm[25], which was reported to be constituted of ~10% human proteins. The metaproteomes of the modern calculus and plaque samples had ~20% of the proteins assigned to human.

To identify similarities and differences between all samples, unsupervised hierarchical clustering of LFQ intensities was performed for all protein entries observed in at least half of the medieval samples (Fig. 2a). The clustering separates the modern samples from the archeological ones and the modern plaque from the modern calculus. Within the medieval samples, two main groups were defined by the cluster analysis, with 16 Tjærby samples falling into Group 1 (G1), and the rest (Tjærby 5, 6, 18, 21, 22, and 23) comprising Group 2 (G2). Subsequent bioinformatic analysis were performed based on these groups. One individual in G2, Tjærby 18, a particularly senescent individual removed from the bioarchaeological analysis (Supplementary Note 2), is also an outlier in terms of his bacterial and human proteome profiles (Figs. 2a and 6, see below).

**Profile of bacterial and other non-eukaryotic proteins**. The majority of the identified proteins were of bacterial origin, of which approximately 90% could be assigned to genus level. In order to assess the relative contribution of different bacterial genera to the total bacterial protein mass, the fractional bacterial genus distribution was calculated by summing all LFQ protein intensities mapped to the taxonomic rank of genus and dividing the result by the total summed protein intensity of all bacterial proteins within each sample (Supplementary Data 1). The top 20 genera found in the samples were visualized as differentially color-coded bar-graphs sorted by the most abundant genera in the medieval samples (Fig. 2b). From this plot, it is evident that the separation of the two groups of medieval individuals is related to the level of abundance of specific genera (Fig. 2b). *Actinomyces* spp., a prominent group of facultative anaerobic Gram-positive bacteria, is the predominant genus in all but one individual. After *Actinomyces* spp., the genera *Olsenella* and *Fretibacterium*, both of which have been implicated in periodontitis[26], are the most abundant in G1. Conversely, G2 is characterized by the presence of oral commensals *Lautropia mirabilis*, *Neisseria* spp., *Streptococcus* spp., and *Cardiobacterium* spp. (Fig. 2b).

A two-tailed *t*-test of summed genera-specific LFQ protein intensities was performed in order to identify significant differentially expressed genera between the two groups in a more global and unbiased manner. The resulting volcano plot shows the differentially expressed genera (Fig. 3). The G1-enriched genera displays significant contributions from *Fretibacterium* spp., *Porphyromonas* spp., *Treponema* spp., *Tannerella* spp., and *Desulfobulbus* sp. oral taxon 041; all of which have been suggested to be involved in clinical periodontitis[26–29]. The presence of *Porphyromonas* spp., *Treponema* spp., and *Tannerella* spp. in the G1 group is of particular interest, since species belonging to these genera are part of the red complex bacteria[30]. The red complex bacteria are known to be strongly associated with periodontal disease and consist of *Treponema denticola*, *Porphyromonas gingivalis*, and *Tannerella forsythia*.

Several virulence factors were identified from these species, and their level of expression are, almost without exception, higher in G1 and absent in modern plaque and calculus (Table 2). For example, the fimbrial proteins identified from the keystone pathogen *P. gingivalis* are critical mediators of initial adhesion

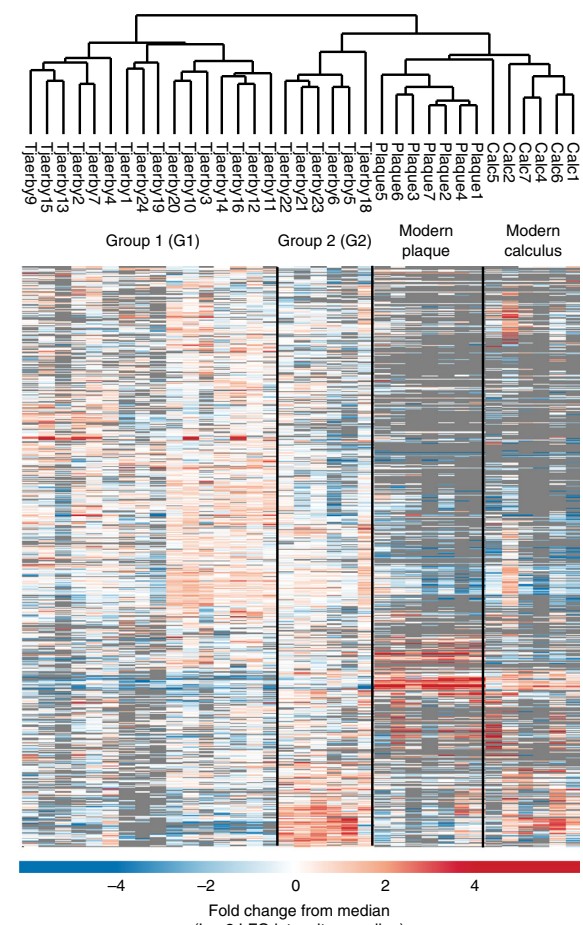

**a**

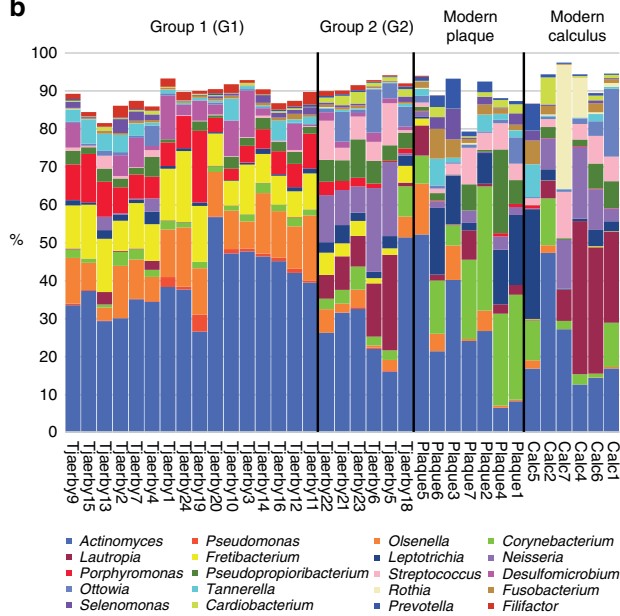

**b**

**Fig. 2** Grouping of samples. **a** Hierarchical clustering showing grouping of samples. **b** Bacterial distribution on genus level based on summed LFQ intensities

and for the invasion of host cells[31], and together with gingipains, they play several roles in pathogenicity. Several proteins from *Methanobrevibacter oralis*, an archaeal genus believed to be an important periodontal disease pathogen[32,33], are also more

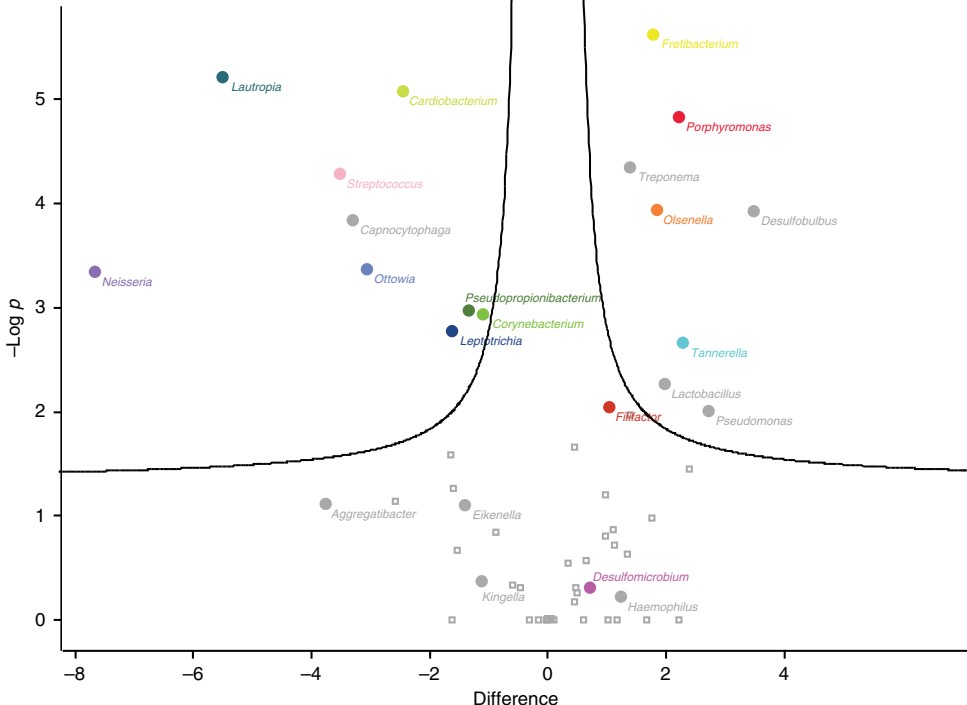

**Fig. 3** Bacterial genera differentially expressed between sample groups. The significantly differentially expressed bacterial genera between G1 (right) and G2 (left) are colored based on the coloring code from Fig. 2. Other interesting genera, not passing the significant threshold are named in the plot

abundant in G1 (Table 2). Although not specific to either group, *Desulfomicrobium orale* is an emerging pathogen of interest[27,34]. *D. orale* is absent in the modern samples and at very low abundance in three of the healthy archeological individuals (Supplementary Data 2). *Lactobacillus* spp., one of the major lactic acid fermenting bacterial genera, is also present with higher abundance in G1.

*Actinomyces* spp., along with *Lautropia mirabilis* and *Neisseria* spp., characterize G2 and are considered to be normal and healthy microbiome commensals[35,36]. *Leptotrichia, Pseudopropionibacterium*, and *Ottowia* are also significant contributors to G2 microbiomes. *Pseudopropionibacterium propionicum* has only recently been identified in ancient dental calculus[37], despite it being a relatively abundant oral inhabitant in G2. Very little is known about *Ottowia* spp.'s physiology and its role in the oral microbiome. Proteins from the *Streptococcus* genus are also overrepresented in G2, and in our dataset this genus mainly consists of the species *S. sanguinis* (Supplementary Data 2), which is able to outcompete the cariogenic species *S. mutans*[38]. Other bacteria of interest detected in G2 include the HACEK group (*Haemophilus* spp., *Aggregatibacter* spp., *Cardiobacterium* spp., *Eikenella corrodens*, and *Kingella* spp.)[39] and *Capnocytophaga* spp. The latter, along with *Aggregatibacter* and *Eikenella corrodens* of HACEK, make up the periopathogenic green complex[30]. *S. sanguinis*, the HACEK group, *Leptotrichia* spp., and *P. pseudopropionibacterium*, are all opportunistic pathogens and have been implicated in conditions such as infective endocarditis[38,40–42].

Both groups contain pathogenic genera and species involved in conditions of the periodontium, but G1 is defined by pathogenic species, whereas G2 is characterized by a number of commensal genera. When comparing the quantitative metaproteome profiles of the two groups with the bioarchaeological analysis, we found no association with biological age, chronological age, size of calculus samples, location of the calculus samples within the mouth or on the tooth, or presence of periapical lesions

(Supplementary Fig. 4). Neither group can be said to be healthy based on the pathological scores (Supplementary Table 1), but the G2 group only has one individual with gross carious lesions compared to nearly half of the individuals in the G1 group. This suggests that the metaproteome profiles to some extent correlate with caries status.

**Profile of human proteins.** In total, 205 human proteins were identified in the medieval samples, of which more than half are known to be extracellular, and a large portion of which are associated with the gene ontology (GO) term 'defense response' (Fig. 4a). Analysis of functional protein-protein interactions among the 205 human proteins using the STRING database[43] reveals two highly interconnected networks represented by proteins with roles in blood coagulation and defense response (Fig. 4b). Among defense response factors, a subgroup of 15 proteins are involved in the acute inflammatory response, including the well-known clinical biomarker for inflammation, C-reactive protein (CRP). When overlapping the human proteins found in the modern plaque, modern calculus, and a previous study[6], we identify 40 out of 43 human proteins previously found in archeological dental calculus (Fig. 5a). The additional proteins we identify are generally of lower abundance, indicating use of a more sensitive MS analysis (Fig. 5b). We find 74 proteins to be unique to the medieval samples and GO-term enrichment of these shows a similar distribution to the overall GO-term enrichment in Fig. 4a. Hierarchical cluster analysis of the two medieval groups shows a set of 13 proteins with higher abundance in G2 (Fig. 6). Many of the G2-enriched proteins, e.g., myeloperoxidase (MPO), lactotransferrin (LTF), neutrophil gelatinase-associated lipocalin (LCN2), and matrix metalloproteinase-9 (MMP9) are specific to neutrophils, a subset of white blood cells that are part of the first line of response to bacterial infection. These same proteins are also observed in the modern healthy samples, indicating a possible normal oral immune response in G2.

**Table 2 Virulence factors and other proteins expression frequency**

| Species | ID | Protein | Log2Ratio G1/G2 | Freq G1 | Freq G2 | Freq plaque | Freq calculus |
|---|---|---|---|---|---|---|---|
| *Porphyromonas gingivalis* | BAK24470.1 | Arginine-specific cysteine proteinase RgpA | 1.56 | 7 | 1 | 0 | 0 |
| | WP_012457306.1 | FimA type I fimbrilin | 0.44 | 14 | 2 | 0 | 0 |
| | BAK25445.1 | FimA type II fimbrilin | 0.45 | 6 | 3 | 0 | 0 |
| | WP_054191322.1 | Fimbrial assembly protein | 4.61 | 10 | 1 | 0 | 0 |
| | WP_005875061.1 | Fimbrillin-A associated anchor protein Mfa1 and Mfa2 | −0.57 | 5 | 1 | 0 | 0 |
| | ALO28935.1 | Major fimbrial subunit protein (FimA) | 0.50 | 2 | 3 | 0 | 0 |
| | Q51826 | Major fimbrium subunit FimA type-3 | 0.32 | 4 | 3 | 0 | 0 |
| | Q51827 | Major fimbrium subunit FimA type-4 | 2.74 | 5 | 1 | 0 | 0 |
| | WP_012457396.1 | Mfa1 fimbrilin | 0.50 | 12 | 6 | 0 | 0 |
| | BAK24619.1 | Mfa1 fimbrilin | 1.25 | 6 | 1 | 0 | 0 |
| | BAK24228.1 | Hemagglutinin protein HagA | 1.22 | 16 | 6 | 0 | 0 |
| | ETA27451.1 | Heme-binding protein | 1.18 | 5 | 2 | 0 | 0 |
| | KXC09143.1 | HmuY protein | + | 4 | 0 | 0 | 0 |
| | BAK25568.1 | Lysine-specific cysteine proteinase Kgp | + | 5 | 0 | 0 | 0 |
| | EOA10826.1 | Outer membrane protein beta-barrel domain protein | + | 4 | 0 | 0 | 0 |
| | WP_005873620.1 | Outer membrane protein 40 | 1.35 | 16 | 5 | 0 | 0 |
| | WP_005873612.1 | Outer membrane protein 41 | 0.59 | 12 | 4 | 0 | 0 |
| | ETA26324.1 | Peptidase | + | 3 | 0 | 0 | 0 |
| | WP_053444556.1 | Peptidase C25 (Gingipain) | 1.50 | 16 | 6 | 0 | 0 |
| | WP_052912324.1 | Peptidase C25 (Gingipain) | 1.26 | 16 | 6 | 0 | 0 |
| *Tannerella forsythia* | NCBIJUET_c_1_62 | Surface layer protein A | 2.39 | 15 | 5 | 2 | 2 |
| | bfor_c_1_1526, NCBIJUET_c_1_63 | Surface layer protein B | 2.01 | 16 | 6 | 6 | 3 |
| | EPF40615.1 | Glycine cleavage system T protein | + | 3 | 0 | 0 | 1 |
| | EGC77611.1 | Glycine reductase complex proprotein GrdE2 | 0.06 | 14 | 6 | 1 | 1 |
| | EPF39880.1 | Glycine/betaine/sarcosine/D-proline reductase family selenoprotein B | 0.07 | 2 | 1 | 0 | 1 |
| | EGC76314.1 | OppA protein | + | 7 | 0 | 0 | 1 |
| *Methanobrevibacter oralis* | mora2671_c_1_73 | Histone | 0.23 | 14 | 6 | 0 | 0 |
| | mora2671_c_10_1468 | Histone | 0.42 | 4 | 1 | 0 | 0 |
| | mora2671_c_19_1855 | Histone | −0.05 | 1 | 1 | 0 | 0 |
| | mora2671_c_6_1057/ mora2671_c_2_473 | Hypothetical | + | 3 | 0 | 0 | 0 |
| | mora2671_c_1_304 | Methyl-coenzyme M reductase subunit gamma | 0.62 | 7 | 2 | 0 | 0 |
| | mora2671_c_16_1762 | NAD(P)-dependent alcohol dehydrogenase | -0.05 | 8 | 2 | 0 | 0 |

Virulence factors from two bacterial species belonging to the red complex, and proteins identified from *Methanobrevibacter oralis*, an archaeal genus believed to be a periodontal disease pathogen. The log2ratio between G1 and G2 indicates a higher abundance of almost all proteins in G2. Frequency in each group is included for transparency of the background data

**Identification of dietary proteins**. Besides the bacterial and human proteins, we also identified three milk proteins: beta-lactoglobulin (BLG) and alpha-S1-casein of bovine origin, and BLG of caprine origin. The caprine BLG is not observed in any of the modern samples, likely reflecting contemporary dairy consumption (Table 3). Based on the identified peptides, almost the full amino acid sequence coverage of BLG was identified from the Tjærby samples. To differentiate between bovine and caprine BLG, we compared their protein sequences and mapped our identified peptides to this multiple sequence alignment (Supplementary Fig. 5). Three tryptic peptides uniquely discriminate between bovine and caprine. However, homolog peptide sequences, whose polymorphism in different species involves asparagine (N) and aspartic acid (D) (marked with asterisks in Supplementary Fig. 5), can hardly be used for species identification in archeological samples. This is due to spontaneous deamidation of N to D during protein degradation, which is known to occur with time. With the current methods, e.g., detection of D in the peptide PTPEG**D**LEILLQK, can equally be interpreted as identification of either the bovine-specific BLG peptide, or of the fully deamidated ovine-specific homolog one.

We also identified two peptides indicating the presence of oat (*Avena sativa*) in four individuals (Supplementary Fig. 6). The identified peptides can be mapped to the protein 12S seed storage globulin from oat (*Avena sativa*), which is a very abundant protein and thus justifies the survival of this protein. The identification indicates a diet containing this nutrient rich cereal.

**Benefits of TMT-multiplexing and offline fractionation**. For five of the medieval samples, sufficient starting material remained to test the possibility of achieving greater proteome depth using a strategy based on TMT-isobaric tag labeling and offline high pH-reversed phase (HpH) fractionation[44]. To keep overall MS measurement time constant, equal peptide amounts were labeled from each of the five samples with a different tandem mass tag (TMT), then mixed, and fractionated offline by HpH and concatenated into 12 fractions. Each of these fractions were analyzed by nanoflow LC–MS/MS using short LC gradients that collectively added up to the same MS time as five individual single-shot runs. With the TMT labelling and fractionation approach, 3359 proteins were identified across the five samples, compared to a

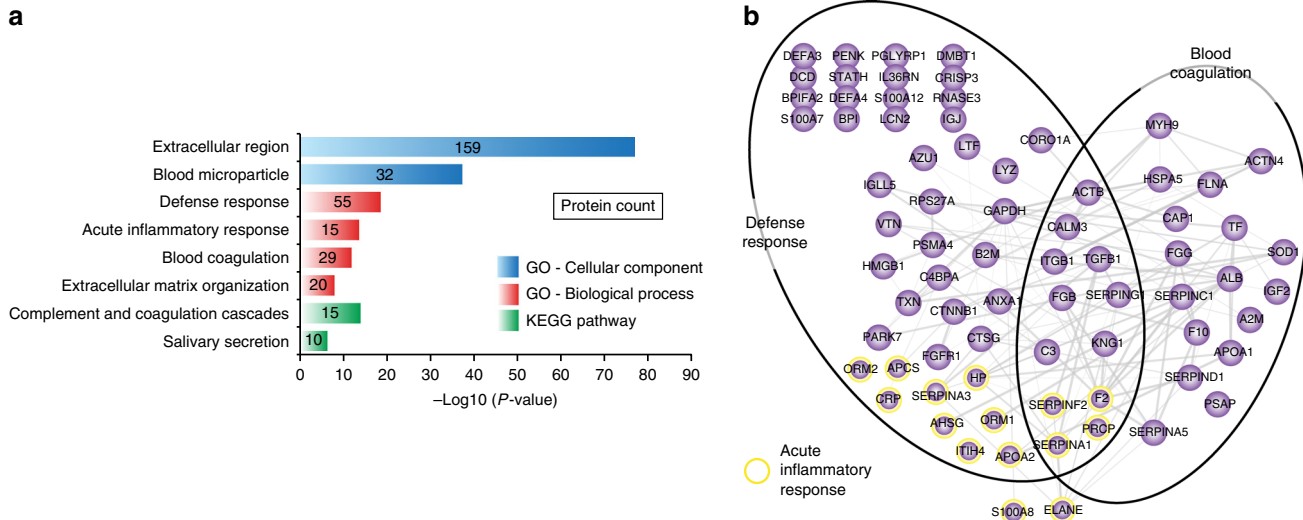

**Fig. 4** Identified human proteins. **a** GO-term enrichment of the human proteins identified. **b** STRING network of selected human proteins with GO annotation

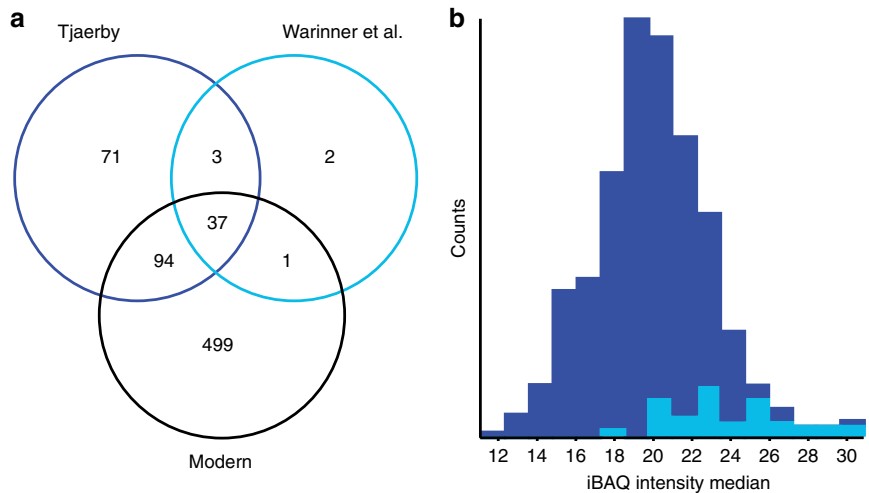

**Fig. 5** Comparison to previous study. **a** Overlap of human proteins identified between medieval, modern and Warinner et al.[6] (previous study). **b** iBAQ intensity of human proteins identified in this study (dark blue) and in Warinner et al.[6] (light blue)

total 2609 from single runs. This represents a gain of more than 30%, and still perfectly correlates with their label-free partners, despite the missing values (Supplementary Fig. 7). With the TMT labeling, we have a minimum of missing values, making quantitative comparison between all samples possible. If we require valid values for just two of the unlabeled single runs, the number of identified proteins markedly decreases from 2609 to 1264, yielding almost three times more quantifiable proteins with the TMT strategy.

## Discussion

The challenges associated with studying and comparing oral microbiome bacteria are mainly due to the individuality of oral flora, differences in sampling strategies and their localities, and the large number of very different microorganisms. Furthermore, many oral bacterial species have only recently been discovered via 16S rRNA studies, and many have proven difficult or even impossible to culture, and thus, remain poorly understood[45,46]. The metaproteomes of dental calculus displays very different bacterial generic profiles compared to a number of clinical

genomics-based studies[47,48]. The curation of comparative DNA/RNA/Protein databases is crucial for meaningful identification of species[37], and it stands to reason that there may be human oral taxa that are no longer present in modern Western populations and, therefore, absent in the core databases. The oral microbiome is highly individual, and even varies within the mouth, as the two samples (#2 and #10) taken from the same individual demonstrate. There is a general agreement that bacterial composition shifts in healthy versus diseased microbiomes leading to dysbiosis[9], but global comparisons across individuals are still complicated to perform due to individuality, and it has even been suggested that microbiomes are as unique as fingerprints[10,49].

However, in the present study we provide, to the best of our knowledge, the deepest quantitative oral metaproteome analyzed across modern and medieval samples to date. We demonstrate how such protein profiles from different species can separate groups of individuals and inform about their general oral health status. We interpret both the number of periopathogenic genera (including *Porphyromonas gingivalis*, *Treponema denticola*, *Tannerella forsythia*, and *Filifactor alocis*) and the presence of the potent virulence factors from the red complex, as indicative of a

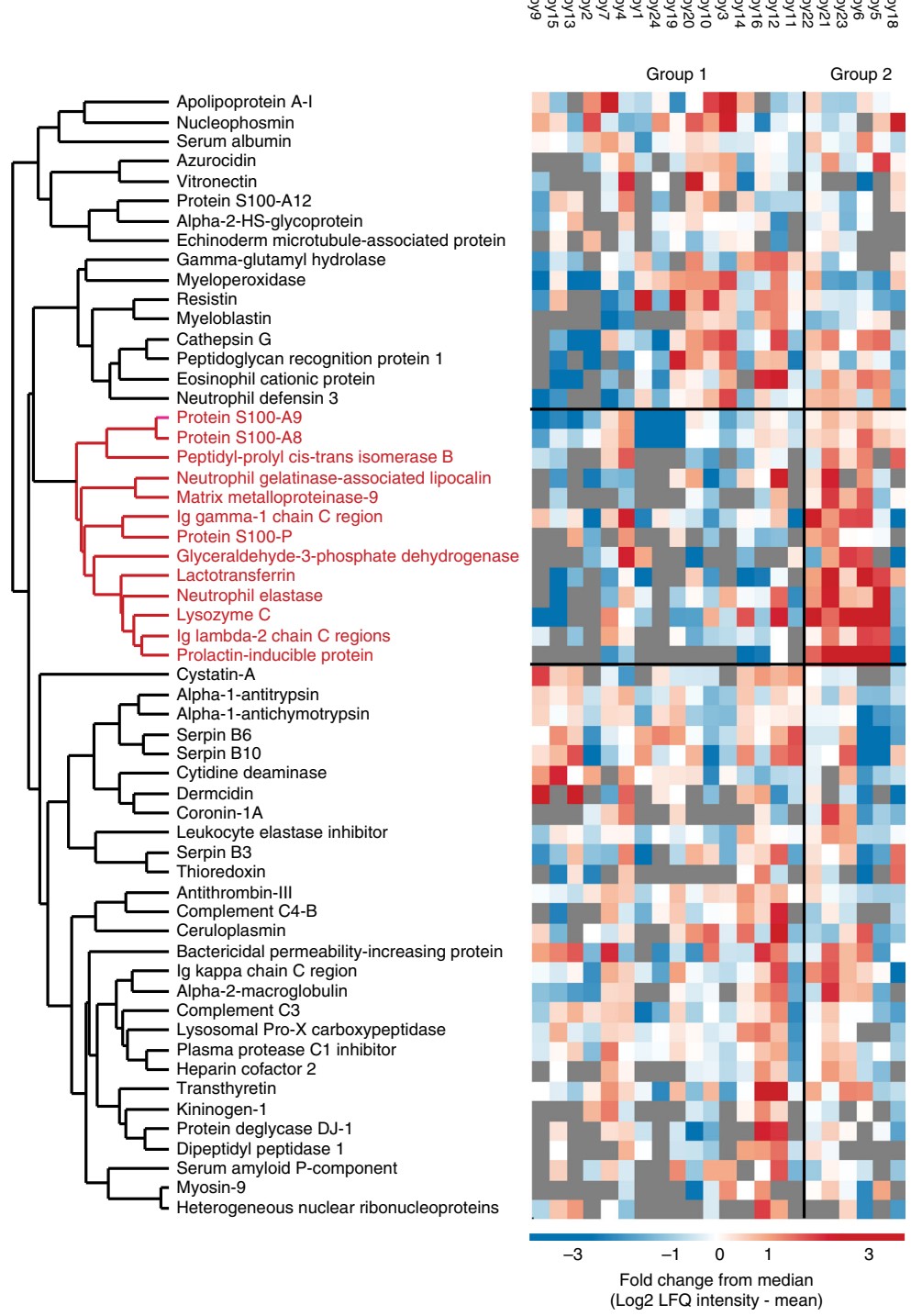

**Fig. 6** Clustering of human proteins. Hierarchical clustering of human proteins (only medieval samples) shows a small subset of proteins that are more abundant in group 2

dysbiotic oral microbiome in the G1 population of individuals. Periodontal disease is a complex polymicrobial infection, and therefore the presence of periopathogens is a risk factor, providing the potential for poor oral health. There are also seven individuals with gross caries in this group. *Streptococcus mutans* and *Streptococcus sobrinus*, the two species considered to be

highly cariogenic[50], are absent from all samples. A genomic study by Belda-Ferre et al.[51] failed to recover *S. mutans* from individuals with carious lesions. However, the genera they suggest correspond to caries are similar to those seen in the bacterial profiles of carious group G1. The specific presence of *Lactobacillus* spp. in this group, and of *S. sanguinis* in G2 group, may be

**Table 3 Number of observation of milk proteins in each sample category**

|  | Tjærby frequency (n = 22) | Plaque frequency (n = 7) | Calculus frequency (n = 6) |
|---|---|---|---|
| BLG—Cow | 20 | 4 | 5 |
| BLG—Sheep/goat | 9 | 0 | 0 |
| Alpha-S1-casein | 3 | 4 | 5 |

BLG Beta-lactoglobulin

risk and preventive factors respectively for dental caries progression.

The core microbiome of G2 is composed of a number of genera that are considered to be normal oral commensals, and this bacterial signature broadly corresponds to the modern microbiome of the healthy Danes analyzed here. The higher presence of HACEK/the green complex in G2, especially *A. actinomycetemcomitans*[27,52], may help to explain why periodontal disease is prevalent in G2 based on visual inspection despite a microbiome that is otherwise suggestive of health. There is no *Desulfomicrobium* in the modern samples and only very low levels of *Fretibacterium*, the former is also at very low levels in three of the healthiest archeological individuals, indicating they may be pathogens of note.

Compared to the modern dental calculus samples, the bacterial profiles of G2 are the most similar, but both medieval groups show a much higher degree of intra-homogeneity than the modern individuals (Supplementary Fig. 8). The homogeneity of ancient calculus could reflect its lifelong deposition, or recovery may be biased towards the proteins that survive degradation. If the former is the case, then we can suggest that oral microbial makeup was largely stable in this medieval population over the centuries. However, we cannot exclude that our observations reflect degradation-robust genera, or the innermost layers of the plaque biofilm less exposed to degrading factors. The changes associated with modern lifestyle, especially the use of antibiotics, oral hygiene, and the highly individual modern diet may also be reflected in the heterogeneity of modern plaque and calculus. As with diet, the role of antibiotic usage and their effect on the oral microbial diversity is poorly understood and requires further research. The heavy use of antibiotics may have profound effect on the structure of the subsequent microbiome formation, especially in early childhood[53,54]. A recent study has also implicated environmental conditions as being an important factor in influencing the population of the oral microbiome[55]. The use of antibiotics and the regular disruption of the microbiome by oral hygiene, compounded by genetics and environmental factors, may explain the very individual nature of modern calculus profiles.

The majority of the human proteins identified in the medieval samples are extracellular and multifunctional in their nature. Many of these proteins represent the first line of defense against microorganisms, and many overlaps with the modern samples. Of the 74 proteins unique to Tjærby, 50 are present in the plasma proteome database, which suggests some degree of blood-contamination in the medieval samples. This would be congruent with individuals with gum bleeding, a common manifestation of periodontal disease. G2 shows a suite of human proteins that are expressed at levels above average compared to G1. These are mainly involved in immune system processes and may reflect a more active and coherent immune response by these individuals. Many of these proteins have antimicrobial functions or are related to inflammation. The clinical biomarker for inflammation,

CRP, shows a larger fractional presence in G1 (4 out of 16). CRP is present in only one individual (Tjærby 23) in G2, which may be explained by the unilateral non-specific florid periosteal new bone on the tibia and fibula. Reactive new bone formation by the periosteum can be suggestive of adjacent injury, inflammation, or similar stimuli[56]. Compared to the seminal study on archeological calculus by Warinner et al.[6], we have almost complete overlap with identification of 40 out the 43 human proteins. Furthermore, we increased the number of identified proteins almost fivefold. This could be due to a combination of sample size, a more sensitive MS method, and a more refined sample preparation. The fractionated TMT-labeled pilot experiment shows great promise for future comparative studies. This strategy largely circumvents the issue of missing values, while increasing the number of overall identifications, which will be beneficial in many projects. The only dietary proteins we identify were from milk and oat.

Based on the metaproteomics data presented here, we are able to use quantitative information to consider the oral microbiome individuality of archeological samples. The results also define two different groups that were not observable in the bioarchaeological analysis, and thus, this method has the potential to add more detailed level of information to bioarchaeological analysis and health reconstruction in past populations. We believe that the different bacterial distributions together with all the other indications from virulence factors, immune-response proteins, and similarity to modern calculus found in the two groups demonstrate different oral health states, a group predisposed to disease, and one to health. Lastly, these results potentially show a shift in the oral microbiome proteome from the medieval period compared to modern samples, and this may reflect changes in lifestyle and contemporary hygiene practices.

## Methods

**Odontological examination of Tjærby individuals**. The Tjærby assemblage is curated by the Retsmedicinsk Institut (Institute of Forensic Medicine), at the University of Copenhagen, and it was from here that samples (n = 22) were collected from 21 osteologically adult male individuals (middle adult (36–45, n = 8) older adult (45+, n = 12), and one individual aged 26–35)[57]. For further details about the sample selection criteria see Supplementary Note 1, and Supplementary Tables 2 and 3. Ordinal scores for every tooth position were recorded for periodontal disease[20], dental calculus[58], periapical lesions[20], and occlusal dental wear[59]. Dental caries were scored from 0 for absent to 3 for gross lesions involving more than half the crown. Antemortem tooth loss was scored when there was extensive or complete remodeling of the alveolus, postmortem tooth loss and/or congenital absence were also noted[60] (See Supplementary Table 1).

**Sample preparation from Tjærby human remains**. Samples were taken by carefully removing the calculus from the tooth with a sterile periodontal scaler and collected in 1.5 mL Protein LoBind Eppendorf tubes (Eppendorf, Germany). The dental tools used for sampling were cleaned with DNA Away (Thermo Fisher Scientific, Denmark) and 70% ethanol. Gloves, foil, and disposable plasticware were replaced after each sampling. The weight of the samples ranged from 15.5 to 145.4 mg (See Supplementary Table 2). All laboratory work on ancient samples was conducted in a dedicated laboratory for human ancient DNA and ancient proteins extraction at the Centre of GeoGenetics, at the Natural History Museum of Denmark.

The calculus samples were demineralized in 1 mL of 15% acetic acid overnight then centrifuged for 10 min. at 2000×g, after which the supernatant was removed. The pellet was then resuspended in lysis buffer (2 M guanidine hydrochloride solution, 10 mM chloroacetamide, 5 mM *tris*(2-carboxyethyl)phosphine) and the pH adjusted with ammonium hydroxide to 7.5–8.5 with the aid of testing strips. The pellet was physically crushed using sterile micro-pestles to ensure maximum coverage of the lysis buffer. Protein denaturation occurred by heating for 10 min at 99 °C, after which the protein concentration was measured by Bradford Assay. Samples with concentrations of 350 µg/mL or more were halved and saved for TMT analysis (see below). Subsequently, samples were digested under agitation at 37 °C for 3 h with 0.2 µg of rLysC (Promega, Sweden) after pH adjustment. The samples were then diluted to a final concentration of 0.6 M guanidine hydrochloride using 25 mM Tris in 10% acetonitrile (ACN). This was followed by overnight digestion with 0.8 µg of trypsin (Promega, Sweden) per sample. To quench the digestion, 10% trifluoroacetic acid (TFA) was added until the pH was <2. The peptides were washed and collected on in-house made C18 StageTips and stored in the freezer until mass spectrometry analysis. Samples were eluted from

the StageTips directly into a 96 well plate with 20 μL of 40% ACN followed by 10 μL of 60% ACN. Samples were evaporated in a SpeedVac™ Concentrator (Thermo Fisher Scientific, Denmark) until ~3 μL was left and 5 μL of 0.1% TFA, 5% ACN was added.

**Ethics.** For plaque and calculus analysis of healthy living individuals, all volunteers gave their written consent to use their data and the project was assessed as not requiring approval by the Copenhagen area science ethics committee, (See Supplementary Note 3); however, all the participants were assigned a random number from 1 to 7 without any reference to personal data for anonymity.

**Plaque sample preparation.** Supragingival plaque samples were collected from the oral surface of the mandibular incisors from seven healthy volunteers (male = 3, female = 4, age ranging from 25 to 35 years with an average of 29 years) by use of a periodontal probe. Samples were collected by a trained dentist (DB), and were immediately deposited in LoBind Eppendorf tubes followed by storage in a −20 °C freezer. The plaque sample amounts were estimated to be 250 μg and were prepared as previously described[61]. Briefly, samples were mixed with lysis buffer (6 M guanidine hydrochloride, 10 mM chloroacetamide, 5 mM tris(2-carboxyethyl) phosphine in 100 mM Tris pH 8.5) and heated for 10 min (99 °C) followed by 4 min. of sonication. Peptide concentrations were determined by NanoDrop (Thermo, Wilmington, DE, USA) measurement. All plaque samples were digested with lysyl endoproteinase (Wako, Osaka, Japan) in a ratio of 1:100 w/w for 3 h. Samples were diluted four times with 25 mM Tris pH 8 to a final concentration of 1.5 M Guanidine hydrochloride and digested overnight with trypsin (modified sequencing grade; Sigma) in a 1:100 w/w ratio.

Digestion was quenched by adding 10% trifluoroacetic acid and centrifuged at 2000×g for 5 min. The resulting soluble peptides in the supernatant were desalted and concentrated on Waters Sep-Pak reversed-phase C₁₈ cartridges (one per sample) and the tryptic peptide mixtures were eluted with 40% acetonitrile (ACN) followed by 60% ACN, and prepared for MS as above.

**Calculus sample preparation.** The modern calculus samples were also collected from the same surfaces as the supragingival plaque samples. Calculus samples were collected with a sterile gracey curette by a trained dentist (DB) and deposited directly in LoBind Eppendorf tubes, followed by storage in a −20 °C freezer. The sample preparation of the modern calculus closely followed that of the Tjærby samples, with weights ranging from 13–35 mg. The main difference was the amount of proteolytic enzymes used. The amount of both rLysC and trypsin added was adjusted to the individual sample protein concentrations, at a 1:100 w/w ratio instead of set amounts. In addition, samples with particularly high concentrations were not fully collected, such that a max of 10 μg of protein would be loaded onto the C18 StageTips. The rest of the sample preparation occurred as above.

**TMT labelling.** Five samples (Tjærby 10, Tjærby 11, Tjærby 12, Tjærby 16, and Tjærby 6) were labelled with TMT (Thermo Scientific). The samples were prepared as described above and after elution from StageTips HEPES was added to a final concentration of 30 mM. TMT reagents were prepared according to manufacturer's protocol and 1.5 μL were added to each 10 μg sample. Samples were incubated for 1 h at room temperature and the labeling reaction was quenched with 1.5 μL 1% hydroxylamine (1:1; TMT:1% hydroxylamine) for 15 min. Samples were pooled and TFA was added to a final concentration of ~1% and pH below 2, and followed by evaporation in a SpeedVac to half volume. The sample was cleaned up on a C18 StageTip.

**HPLC fractionation.** After labelling and pooling, the sample was evaporated to around 5 μL in a SpeedVac™ Concentrator and then rehydrated to 12 μL with 25 mM ammonium bicarbonate (ABC). The fractionation was performed using a Waters ACQUITY UPLC Peptide CSH C18 1.7 μm, 1 × 150 column on an Ulti-mateMicro HPLC (Dionex, Sunnyvale, CA, USA). The sample was loaded onto the column and then eluted at a rate of 30 μL/min. Twelve fractions were collected using a Dionex AFC-3000 fraction collector in a 96 well plate. The fractionation well was changed every minute between the 12 wells starting from minute 4. Buffer A was 5 mM ABC and Buffer B was 100% acetonitrile (ACN). The separation gradient was: 4.5 to 22.5% Buffer B at 50 min, then to 63% over the next 4 min and held constant for 6 min to a total run-time of 60 min. After this, fractionation was stopped and for 2 min, Buffer B was increased to 81% and held for 8 min before dropping back down to 4.5% to proceed with washing and equilibrating the column. The fractions were acidified with formic acid to get rid of carbonate and dried down completely in the SpeedVac. The samples were reconstituted in 50 μL of 80% ACN and 0.1% TFA, and evaporated in SpeedVac™ Concentrator until ~5 μL was left and resuspended, as above.

**LC-MS.** LC–MS/MS setup for unfractionated samples was as described in Demarchi et al.[62] (Copenhagen setup). In short, the samples were separated on a 50 cm PicoFrit column (75 μm inner diameter) in-house packed with 1.9 μm C₁₈ beads (Reprosil-AQ Pur, Dr. Maisch) on an EASY-nLC 1000 system connected to a

Q-Exactive HF (Thermo Scientific, Bremen, Germany). The peptides were separated with a 165 min. gradient.

The Q-Exactive HF was operated in data-dependent top 10 mode. Full scan mass spectra were recorded at a resolution of 120,000 at m/z 200 over the m/z range 300–1750 with a target value of $3 \times 10^6$ and a maximum injection time of 20 ms. HCD-generated product ions were recorded with a maximum ion injection time set to 108 ms through a target value set to $2 \times 10^5$ and recorded at a resolution of 60,000.

TMT-fractionated samples were separated on a 15 cm column (75 μm inner diameter) in-house laser pulled and packed with 1.9 μm C₁₈ beads (Reprosil-AQ Pur, Dr. Maisch) on an EASY-nLC 1000 system connected to a Q-Exactive HF (Thermo Scientific, Bremen, Germany). The column temperature was maintained at 40 °C using an integrated column oven (PRSO-V1; Sonation GmbH, Biberach, Germany). The peptides were separated with an 80 min gradient with increasing buffer B (80% ACN and 0.1% formic acid), going from 3 to 10% in 5 min, 10 to 30% in 60 min, 30 to 80% in 10 min followed by a 5 min wash and re-equilibrating step. All these steps were performed at a flow rate of 250 nL/min.

The Q-Exactive HF instrument (Thermo Scientific, Bremen, Germany) was run in a data-dependent acquisition mode using a top 10 Higher-Collisional Dissociation (HCD)–MS/MS method with the following settings. Spray voltage was set to 2 kV, S-lens RF level at 50, and heated capillary at 275 °C. Full scan resolutions were set to 120,000 at m/z 200 and the scan target was $3 \times 10^6$ with a maximum fill time of 20 ms. Target value for HCD–MS/MS scans was set at $2 \times 10^5$ with a resolution of 30,000 and a maximum fill time of 60 ms. Normalized collision energy was set at 33 and the isolation window was 0.8 m/z.

**Data analysis.** Raw files were processed with MaxQuant version 1.5.3.36[22] using default settings and oxidation (M), Acetyl (protein N-term), deamidation (NQ), Q ->pyro-E, E->pyro-E and hydroxyproline was set as a variable modification and carbamidomethyl (C) as fixed modification. Digestion enzyme was trypsin with maximum two missed cleavages. The minimum score of modified and unmodified peptides was set to 40. Data were searched against a concatenated FASTA file, consisting of the human reference proteome from UniProt, entire SwissProt[24], and the Human Oral Microbiome Database (HOMD)[23], retrieved August 2014 without applying FDR cutoff. The aim was to increase peptide and protein identifications, while controlling false positives in a conservative manner. Therefore, we stratified the search space into three uneven groups, namely human, bacteria, and other (consisting of all other taxa such as food remains and Archaea). FDR calculations were performed separately within each of these groups at the peptide level using the peptides.txt file from MaxQuant output. The FDR was calculated analogous to Cox and Mann[22] as follows. In order to determine a cutoff score for a specific FDR, all peptide identifications—from the forward and the reverse database—were sorted by their Andromeda-score in descending order. Peptides were accepted until 1% of reverse hits/forward hits had accumulated.

Furthermore, the resulting peptides were quality control filtered based on the following criteria. Entries where the Leading razor protein is a reverse protein hit and there is a valid entry (not a missing value) in the Proteins column were removed. In order to work with MaxQuant's LFQ intensities we used the remaining Leading razor proteins identifiers from the peptides.txt to map and filter the proteinGroups.txt file through the Protein IDs column. The proteinGroups.txt was then filtered by removing all protein-group entries with a value of less than two Razor + unique peptides per raw file.

Lowest Common Ancestor (LCA) searches were performed as follows[63]. Accession numbers from the majority protein IDs column in the proteinGroups.txt were used to retrieve information about LCA for each protein-group entry. To find the LCA of a protein group, accession numbers with the most peptide-associations were selected and mapped to species and their full taxonomic lineage. The lowest taxonomic rank of the intersection of the latter yielded the LCA. All LCA searches resulting in the parvorder Catarrhini (primates) were set to be human including entries in the other category that had a human accession number in the protein group.

The final resulting protein-group file was manually filtered for reverse hits and common contaminants. The species/genus assignment from the LCA was manually validated in the other category and reassigned when needed. Collagen and keratin were not considered in this study.

After the manual filtering of the protein-group file, a new peptide file was generated by mapping Majority protein IDs to Proteins in the all-Peptides file. The Unique sequence column in this file was used for counting peptides in Table 3 and LFQ entries in protein-group file were used for protein quantitation.

Further data analysis was done in Perseus version 1.5.2.6 and 1.5.1.12[64]. For overall hierarchical clustering using Pearson correlation distances (Fig. 2a), only entries observed in minimum half (11) of the medieval samples were used. LFQ intensities were log2-transformed and normalized by subtraction of the median.

For bacterial distribution (Fig. 2b), the summed LFQ intensities per genera as a fraction of total bacterial LFQ intensity per raw file were used. A two-tailed t-test was done on log2-transformed fraction values to determine significant abundance differences of bacterial genera in the two medieval sample groups (Fig. 3).

Bray–Curtis dissimilarity was calculated using Python version 3.6 in conjunction with scikit-bio version 0.5.2 [http://scikit-bio.org]. The results in Supplementary Data 1 were used as input for the calculations. Percent values were

scaled up by multiplying by $10^6$ and decimals cut in order to transform the percent values to integers.

**TMT data analysis**. The raw files from the TMT experiment were processed with MaxQuant version MaxQuant_1.5.5.4i with the same parameter settings as described above except that the parameter type in the group-specific pane was set to Reporter ion MS2 and label to 6plex TMT. The resulting output files were post-processed as described above.

## Data availability

The mass spectrometry proteomics data have been deposited to the ProteomeXchange Consortium via the PRIDE[65] partner repository with the dataset identifier PXD008601. All other data supporting the findings of this study are available from the corresponding authors on reasonable request.

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

## Acknowledgements

This work was in part funded by the University of Copenhagen (KU2016 programme), as well as Danmarks Grundforskningsfond (Danish National Research Foundation (DNRF128)). Work at Novo Nordisk Foundation Center for Protein Research (CPR) is funded in part by a generous donation from the Novo Nordisk Foundation (Grant number NNF14CC0001). E.C. is supported by an "Experiment" grant (17649) from VILLUM Fonden. The funders had no role in study design, data collection and analysis, decision to publish, or preparation of the manuscript. Museum Østjylland, Lutz Klassen and Chief Curator Ernst Stidsing are kindly thanked for allowing sampling from the Tjærby skeletons and for permission to use an excavation photo. Jan Refsgaard and Stephanie Munk are thanked for bioinformatic input and graphic design, respectively.

## Author contributions

E.W., E.C, N.L. and J.V.O. initiated the project. L.T.L. carried out bioarchaeological analysis. L.T.L., M.M., R.R.J.-C. and A.K.F. collected and prepared samples. D.B. collected modern samples and provided critical input for the manuscript. R.R.J.-C. and D.L. carried out mass spectrometry data analysis. C.D.K. and L.J.J. provided supervision of data analysis. E.C. and J.V.O. supervised the interpretation of the results and the formulation of the conclusions. R.R.J-C. and L.T.L. wrote the manuscript and all authors reviewed and approved it.

## Additional information

**Competing interests:** The authors declare no competing interests.

