## [Peer Review File · Nature Communications]

Reviewers' Comments:

Reviewer #1:

Remarks to the Author:

In this manuscript, Jersie-Christensen et al. characterize and quantify the metaproteome of dental calculus from a medieval cemetery in Denmark using state-of-the-art mass spectrometry and label free and TMT quantification techniques. This studies shows the power of proteomics to understand past human populations and better understand oral disease in these populations. This is the first application of TMT to archeological remains, but the key points is the excellent usage of label free methods to quantify the metaproteome of these individuals.

The statistics applied in this paper are strong and applied in a way to allow for easy interpretation of these complex results.

My one major concern is the limited nature of the ethics letter for human subjects. It may be the result of my experience with human subjects research, but it is unclear why the living human samples were approved/did not require approval. Typically IRB approval is required for any study that could have identifiable information about a living human regardless of the type of study. Please describe how or if the volunteers were anonymized, and also how they were chosen/recruited. If this approval is clarified/added, I think this is an excellent study that has interesting implications for future research in this area.

Minor comments:

Page 7 line 117: How do you account for the statistical scoring penalties as a result of a database of this size? You mention separately applying FDR to each group, but your scores will be worse using a huge concatenated database. Please clarify.

Page 9 Line 144: No figure 4d is present. Please clarify.

Page 10 Line 165: It appears that Filifactor is insignificant based on the curves on the volcano plot, but listed as significant on lines 163-165. Please clarify this point. Additionally, on Fig. 3 why are some of the genera that look significant colored gray like the insignificant taxa?

Page 10 Line 174: "Several proteins from *Methanobrevibacter oralis*, an archaeal genus believed to be an important periodontal disease pathogen" I don't see this species on fig 3. Please clarify that Fig. 3 is only bacterial genera or this archael species was accidentally excluded.

Page 12 Line 218: Please define what proteins MPO, LTF, LCN2, and MMP9 gene names represent.
Page 18 Line 330: "oats (*Avena sativa*), but this was rejected after manual inspection of the raw data" I'm not convinced that rejection of the oat peptides was warranted based strictly on the proline effect. Supplementary figure 1a has an extensive y-ion sequence tag supporting its identification and NCBI blastp of that sequence tag with the exclusion of the ALP has a strong match to oat. I would be more convinced to reject these peptides if the fragmentation was sparse or if it was basically only a match based on MS1, but looking at the annotated spectra, I remain unconvinced that they should be excluded.

Page 20 Line 346: Was there a reason why females were not chosen for this study? Additionally, in this section please list what museum houses these skeletal remains. And for clarity, are the A codes in Table 1 the accession numbers for each of these individuals? If so, please clarify this point in this section. If not, please list what the museum accession numbers for each individual, for future reproducibility.

Page 21 Line 381: When approval for the living humans is clarified, please add that the sampling was approved by the ethics board or IRB (whichever applies).

Page 36 Table 1: The line weights of the diagrams are a little hard to see. Please increase them (even a small amount will help). Also, please add a column listing the group each individual ended up in during the clustering analysis.

Supplementary: In the supplement, if permission is given, can you provide a few photographs showing how the periodontal troughs were measured (especially a deep one), examples of the dental caries, and maybe an example of the abscesses? I think it will help clarify your text descriptions of each individual. Also an example of one of the --- individuals to show how different the teeth are.

Reviewer #2:

Remarks to the Author:

Jersie-Christensen et al. performed a metaproteomics study on the dental calculus samples collected from 21 archaeological (medieval) individuals as well as 7 modern healthy living individuals. They found that the metaproteome profiles of both bacterial and human proteins clearly separate the archaeological individuals into two groups with one group (G2) clustering close to modern healthy individuals; they found no association between the clustering and the physiological/pathological status as measured with bioarchaeological analysis. They further identified the differentially abundant host, bacterial proteins and microbial species between the two subgroups and suggested that the G2 individuals were characterized by the presence of oral commensals, while G1 individuals were characterized by the potent pathogens. This study provides a good dataset for understanding the oral microbiome functions of ancient population, however, this manuscript generally contains too much over speculated discussions and over interpreted conclusions, in particular those relating to healthy status. There is also a lack of sufficient statistical evaluation for most of the key observations. Therefore, I do not recommend for publication in Nature Communications.

Major concerns

1. Using unsupervised hierarchical clustering of protein LFQ intensity, the authors observed four clusters including modern plaque cluster, modern calculus cluster and two medieval clusters (G1 and G2). Is this clustering observation surprising? As indicated in the "Methods" section, the protein extraction and digestion methods are different for ancient and modern samples. Could that severely affect the sample clustering? The sum LFQ intensity of the samples varies >3 fold (as calculated from the database search output file: proteinGroups_workfile.txt). It seems the clustering also associated with the sum LFQ intensity variations. In addition, if I am correct, the raw file names indicate that the samples from different groups were analyzed on mass spectrometer as different batches. Whether the batch effects were evaluated? Could the clustering also be associated with the MS date?
2. The authors tried to link the oral metaproteome variations to oral health status in ancient individuals and made thorough discussions on the potential roles of G2-enriched commensals or G1-enriched potential pathogens. To further support the results, the calculus samples collected from healthy living individuals were also included for comparison. Why healthy individuals were used for the comparison, instead of periodontitis patients? It will be more convincing to also include a group of periodontitis patients since all the ancient samples displayed evidence of periodontitis (as indicated in Lines 66-68).
3. Lines 293-294: "The modern plaque appears to be more variable than the modern calculus, possibly representing the dynamic nature of the ...". It is not obvious to see the differences from the hierarchical clustering tree in Figure 2a. Supporting statistical analysis, such as the statistical comparison of the inter-individual distances, should be provided to make the above claims.

Minor comments:

1. Line 144: fig. 4d \diamond Fig. 6
2. Figure 6: a set of 13 proteins with higher abundance in G2 compared to G1 were indicated, including those normal oral immune-related proteins specific to neutrophils. Are these proteins

also higher abundance in the modern healthy samples? Why the modern sample groups were not included in the heat map?

3. The TMT results seem to be distracting since the whole manuscript focus on the differences between the two subgroups of ancient calculus samples and their comparisons with samples collected from healthy living individuals.

4. Figure 8: sample names should be labelled instead of the TMT channel names. More description in figure legend is needed. For example, how the fold changes were calculated (median was calculated by rows or columns)? Why is TMT131 obviously different with others?

5. Details of the 7 healthy individuals are needed, such as the age and gender. Are they matched with the archaeological individuals?

Reviewer #3:

Remarks to the Author:

This study presents the results on dental calculus metaproteomes from 21 humans archaeologically recovered from a medieval (ca. 1100-1450 CE) cemetery at Tjaerby, Denmark, and constitutes the first attempt of quantitative metaproteomics of archaeological samples, following a seminal study by Warinner et al (2015). The authors state that metaproteomics has the potential to add new and quantitative levels of molecular information on the oral health status of individuals recovered from archaeological contexts. I was asked by the journal editors to comment on the bioarchaeological aspects of this study, and therefore the comments that follow have this domain as their sole focus.

The authors present novel results, of interest not only in assessing oral health in the specific population studied (medieval, ca. 1100-1450 CE cemetery at Tjaerby, Denmark), but also of clear methodological interest, both in terms of approaching oral health status of past individuals, but potentially also for investigating dietary proteins and milk consumption in particular, through the differentiation between bovine and caprine milk proteins found in calculus. As such, the results are of interest to the bioarchaeological research communities. While the work is convincing, some further information on the following aspects would improve the paper, its potential impact, and reproducibility:

(1) Why were only male individuals included in the study? While there may be a good reason for this, it is currently not stated explicitly in the paper. This needs to be discussed. What is also not clear is whether the authors considered (or will consider) making similar analyses for females, and whether there are calculus deposits which would allow subadults to be analysed.

(2) Why does the study include only middle/older adults, save for one younger adult individual? Again, there may be a good reason for this (e.g. insufficient calculus deposits?), but this needs to be discussed explicitly in the paper.

(3) As to reproducibility of this kind of work, it would be very useful to have more information on the calculus deposits. The bioarchaeological reader would benefit from a fuller description (e.g. in the supplementary data) of the calculus deposits in each dentition, their location and character (e.g. supra- or subgingival), as well as extent. At minimum, at least one illustration of a calculus deposit could be included, e.g. within the sampling protocol figure, or within the supplementary materials. Further, the choices as to where an individual's calculus deposits were (preferentially) sampled from is not clear.

(4) Photograph/s of a representative dentition and alveoli showing periodontal disease within this population would be useful. If the maximum number of illustrations has already been reached, such an image could be included within the sampling protocol figure as one of the sub-figures.

(5) The importance of the medieval site is not elaborated sufficiently for the reader to get a sense of why this medieval parish site has any importance beyond a mere case study. Why should we learn more about the oral health of a medieval village population in Denmark? What are the larger (archaeological/human health) questions here? What is the impact of the results derived from this particular site – or could any assemblage with individuals with sufficient amount of calculus have been chosen? The authors state that '[t]he site is of great interest because it has been fully

excavated, and the skeletons reflect a typical, medieval village parish population', but do not qualify this statement by giving information on how many (contemporary) sites in Denmark have been completely excavated, and whether any other medieval village parish populations are available for study. An explicit statement on why the study of oral health within a typical, medieval village parish population is of importance would be welcome. Given that only a relatively small number of skeletons (n=21, all male) from the overall population was analysed it may be of lesser importance that the site was fully excavated. The paper needs to elaborate more fully on the importance/uniqueness of the site and/or the skeletal population/assemblage, and the wider questions that can be explored through them, or state that the analyses could have been conducted on any other (contemporary) skeletal assemblage to illustrate the potential of the method.

The approach taken by the paper is potentially of great interest to the bioarchaeological community, and may influence how we approach past oral health in the future.

Kirsi O. Lorentz

Point-by-point rebuttal to reviewer comments for NCOMMS-18-03009

“Quantitative metaproteomics of medieval dental calculus reveals individual oral health status” by Jersie-Christensen et al.

Our responses to each of the reviewer points are indicated *italicized blue text*.

New text added to the manuscript is indicated in quotation marks below responses.

Reviewers comments:

Reviewer #1:

In this manuscript, Jersie-Christensen et al. characterize and quantify the metaproteome of dental calculus from a medieval cemetery in Denmark using state-of-the-art mass spectrometry and label free and TMT quantification techniques. This studies shows the power of proteomics to understand past human populations and better understand oral disease in these populations. This is the first application of TMT to archeological remains, but the key points is the excellent usage of label free methods to quantify the metaproteome of these individuals.

The statistics applied in this paper are strong and applied in a way to allow for easy interpretation of these complex results.

My one major concern is the limited nature of the ethics letter for human subjects. It may be the result of my experience with human subjects research, but it is unclear why the living human samples were approved/did not require approval. Typically IRB approval is required for any study that could have identifiable information about a living human regardless of the type of study. Please describe how or if the volunteers were anonymized, and also how they were chosen/recruited. If this approval is clarified/added, I think this is an excellent study that has interesting implications for future research in this area.

Thank you for pointing our attention to such a delicate and important aspect. We contacted the regional science ethics committee in the Copenhagen area and the chief consultant assessed that the study did not require notification to the ethics committee since it was considered a method study. The 7 healthy volunteers were recruited among the employees of the Novo Nordisk Foundation Center for Protein Research and assigned a random number from 1 to 7, without any reference to personal data. We have changed the ethics section to the following and hope that this clarification satisfies your concern:

“All volunteers gave their written consent to use their data and the project did not require approval by an ethics committee (Supplementary Information 2, in Danish), however, all the participants were assigned a random number from 1 to 7 without any reference to personal data for anonymity”.

Minor comments:

Page 7 line 117: How do you account for the statistical scoring penalties as a result of a database of this size? You mention separately applying FDR to each group, but your scores will be worse using a huge concatenated database. Please clarify.

Thank you for giving us the opportunity to clarify this point. We expect the Posterior Error Probability (PEP) values of peptides and protein groups to be negatively affected in meta-proteomics experiments by greatly increasing the search space (protein database size), specifically in regard to “non-sense” sequences that do not occur in the sample. Since for such cases it will be equally likely that the top hit of a Peptide Spectrum Match (PSM) based on PEP values is assigned to the decoy or to the target search space. On the contrary, we assume the Andromeda scores to be invariant, since such a score is calculated by comparing an experimental to an in-silico spectrum and therefore constant, regardless of the relative ranks of PSMs due to small or large search spaces. We use the Andromeda score to select the best scoring PSM and calculate the FDR within each of the 3 aforementioned groups, by simply sorting from highest to lowest scoring hit, and including all hits until 1% decoy hits have accumulated.

Page 9 Line 144: No figure 4d is present. Please clarify.

Thank you for spotting this inconsistency. The text has been edited from: “(Fig. 2a and fig. 4d)” to: “(Fig. 2a and fig. 6)”.

Page 10 Line 165: It appears that *Filifactor* is insignificant based on the curves on the volcano plot, but listed as significant on lines 163-165. Please clarify this point. Additionally, on Fig. 3 why are some of the genera that look significant colored gray like the insignificant taxa?

Thank you for this observation, Filifactor has now been removed from the text.

The paragraph:

*“The G1-enriched genera displays significant contributions from *Fretibacterium* spp., *Porphyromonas* spp., *Treponema* spp., *Tannerella* spp., and *Desulfobulbus* sp. oral taxon 041, and ***Filifactor alocis***; all of which have been suggested to be involved in clinical periodontitis²³⁻²⁶.”, *has been changed to:**

*“The G1-enriched genera displays significant contributions from *Fretibacterium* spp., *Porphyromonas* spp., *Treponema* spp., *Tannerella* spp., and *Desulfobulbus* sp. oral taxon 041; all of which have been suggested to be involved in clinical periodontitis²³⁻²⁶.”*

The coloring in figure 3 is based on the coloring in figure 2.

The figure text has been changed from:

“Significantly differentially expressed genera between G1 (right) and G2 (left) are colored (except *Filifactor*).”, *to:*

The significantly differentially expressed **bacterial** genera between G1 (right) and G2 (left) are colored **based on the coloring code from figure 2.**”

Page 10 Line 174: “Several proteins from *Methanobrevibacter oralis*, an archaeal genus believed to be an important periodontal disease pathogen” I don’t see this species on fig 3. Please clarify that Fig. 3 is only bacterial genera or this archaeal species was accidentally excluded.

Thank you for pointing this out. The protein information is found in table 3 and the figure text for figure 3 has been changed from:

“Significantly differentially expressed genera between G1 (right) and G2 (left) are colored (except *Filifactor*).” *to:*

The significantly differentially expressed **bacterial** genera between G1 (right) and G2 (left) are colored **based on the coloring code from figure 2.**”

Page 12 Line 218: Please define what proteins MPO, LTF, LCN2, and MMP9 gene names represent.

Thank you for catching this obvious mistake. The text has been changed from:

“Many of the G2-enriched proteins (e.g. MPO, LTF, LCN2, and MMP9) are specific to neutrophils..” *to:*

“Many of the G2-enriched proteins (**e.g. myeloperoxidase (MPO), lactotransferrin (LTF), neutrophil gelatinase-associated lipocalin (LCN2), and matrix metalloproteinase-9 (MMP9)**) are specific to neutrophils.”

Page 18 Line 330: “oats (*Avena sativa*), but this was rejected after manual inspection of the raw data” I’m not convinced that rejection of the oat peptides was warranted based strictly on the proline effect. Supplementary figure 1a has an extensive y-ion sequence tag supporting its identification and NCBI blastp of that sequence tag with the exclusion of the ALP has a strong match to oat. I would be more convinced to reject these peptides if the fragmentation was sparse or if it was basically only a match based on MS1, but looking at the annotated spectra, I remain unconvinced that they should be excluded.

*Thank you for this comment. We discussed this issue a lot and agreed that rather than making conclusions on this piece of evidence we would include it in the manuscript for the reader themselves to make the conclusion. Now, after your review, we fully agree with your interpretation of the high-quality of the MS/MS spectra identifying *Avena sativa* and we therefore decided to include it in the manuscript as an identification. In the Dietary proteins result section we have now included the following paragraph.*

“We also identified two peptides indicating the presence of oat (*Avena sativa*) (Supplementary Figure 2a and b). The identified peptides can be mapped to the protein 12S seed storage globulin from oat (*Avena sativa*), which is a very abundant protein and thus justifies the survival of this protein. The identification indicates a diet containing this nutrient rich cereal.”

Page 20 Line 346: Was there a reason why females were not chosen for this study? Additionally, in this section please list what museum houses these skeletal remains. And for clarity, are the A codes in Table 1 the accession numbers for each of these individuals? If so, please clarify this point in this section. If not, please list what the museum accession numbers for each individual, for future reproducibility.

The following information about why only older males were selected has been added to the introduction:

“In order to look at the diversity in microbiome species, as well as disease biomarkers from dental calculus, we were interested in individuals most likely to have osseous changes and destruction of the alveolar bone related to periodontitis. Older males were selected based on the understanding that males often have a more aggressive inflammatory immune response compared to females¹⁵⁻¹⁷. Males over 45 with a maxilla, mandible, teeth, evidence of periodontal disease, and sufficient dental calculus for sampling, were the initial criteria for inclusion. However, this yielded only 12 samples. By adding younger males, it was possible to collect 22 dental calculus samples from 21 individuals.”

For clarification of A numbers in Table 1 the header of the first column has been changed from:

“Sample”

to:

“Sample/

Accession number for each individual”

See Table 1 on next page.

In the main text, curatorial details have been included. The following text has been added to the manuscript in the material and method section:

“The Tjærby assemblage is curated by the Retsmedicinsk Institut (Institute of Forensic Medicine), at the University of Copenhagen, and it was from here that...”

Page 21 Line 381: When approval for the living humans is clarified, please add that the sampling was approved by the ethics board or IRB (whichever applies).

Thank you for your concern regarding the ethics, and we refer you to our previous clarification:

We contacted the regional science ethics committee and the chief consultant assessed that the study did not require notification to the ethics committee since it was considered a method study.

The 7 healthy volunteers were recruited among the employees of the Novo Nordisk Foundation Center for Protein Research and assigned a random number from 1 to 7 without any reference to personal data.

We have changed the ethics section to the following and hope that this clarification satisfy your concern.

“All volunteers gave their written consent to use their data and the project did not require approval by an ethics committee (Supplementary Information 2, in Danish), however all the participants were assigned a random number from 1 to 7 without any reference to personal data for anonymity”

Page 36 Table 1: The line weights of the diagrams are a little hard to see. Please increase them (even a small amount will help). Also, please add a column listing the group each individual ended up in during the clustering analysis.

Supplementary: In the supplement, if permission is given, can you provide a few photographs showing how the periodontal troughs were measured (especially a deep one), examples of the dental caries, and maybe an example of the abscesses? I think it will help clarify your text descriptions of each individual. Also an example of one of the --- individuals to show how different the teeth are.

Good point, thank you, the line weights have been increased for visibility. Clarification of the groupings has also been made. Three photographs from the bioarchaeological analysis have been added to the supplementary information (1D, 1E, and 1F) showing, periodontal troughs, a carious lesion, and an abscess.

Sample/ Accession number for each individual	Gross caries 	Teeth lost antemortem 	Presence of periodontal troughs 	Total Score
#20 A1235	-	+	+	- + +
#2 / 10 A1408	-	+	+	- + +
#11 A1442	-	+	+	- + +
#9 A1294	+	+	-	+ + -

#12 A1623	+	-	+	+ - +
#3 A1416	+	-	+	+ - +
#7 A1899	+	-	+	+ - +
#19 A970	+	-	+	+ - +
#14 A1764	+	-	-	+ - -
#1 A795	+	-	-	+ - -
#4 A1637	-	-	+	- - +
#24 A1968	-	-	-	- - -
#15 A1893	-	-	-	- - -
#13 A1635	-	-	-	- - -
#16 A1898	-	-	-	- - -
#18* A983	N/A	N/A	N/A	N/A
#21 A1453	+	+	+	+ + +
#5 A1671	-	-	+	- - +
#6 A1673	-	-	+	- - +
#22 A1866	-	+	-	- + -
#23 A1664	-	-	-	- - -

Table 1; Pathology Scores based on the presence of gross caries, >2 teeth lost antemortem, and the presence of periodontal troughs, from 'unhealthiest' with three positive scores to 'healthiest' with three negative scores (see also Supplementary Table 1). **The bold line separates Group 1 from Group 2.**

Supplementary 1D) The photograph of #6 (A1673) showing periodontal troughs along the buccal edge of the left molars (note also glue in the socket of LL5).

Supplementary 1E) A large carious lesion that has destroyed the crown of the upper right first molar with associated abscess formation in individual #7 (A1899)

Supplementary 1F) Large abscess corresponding to the upper right first molar of #11 (A1442).

Reviewer #2:

Jersie-Christensen et al. performed a metaproteomics study on the dental calculus samples collected from 21 archaeological (medieval) individuals as well as 7 modern healthy living individuals. They found that the metaproteome profiles of both bacterial and human proteins clearly separate the archaeological individuals into two groups with one group (G2) clustering close to modern healthy individuals; they found no association between the clustering and the physiological/pathological status as measured with bioarchaeological analysis. They further identified the differentially abundant host, bacterial proteins and microbial species between the two subgroups and suggested that the G2 individuals were characterized by the presence of oral commensals, while G1 individuals were characterized by the potent pathogens. This study provides a good dataset for understanding the oral microbiome functions of ancient population, however, this manuscript generally contains too much over speculated discussions and over interpreted conclusions, in particular those relating to healthy status. There is also a lack of sufficient statistical evaluation for most of the key observations. Therefore, I do not recommend for publication in Nature Communications.

We thank reviewer #2 for his/her comments, thorough reading, and inspection of our data, and were happy to read that the reviewer found our study to provide a good dataset for understanding the oral microbiome functions of ancient population. From the comments and criticism of reviewer #2 it is clearly evident to us that reviewer #2, just like reviewer #1, has an extended experience in reviewing proteomics studies. However, reviewer #2 is likely less familiar with palaeoproteomics studies. Therefore, before we start responding point by point to the valuable comments from reviewer #2, we believe it is worth sharing a few general considerations about which principles should be considered when evaluating the robustness of results based on ancient proteomics. We agree that publication of claims based on palaeoproteomics evidence should fulfill several high-quality criteria. In our now more than 10 years-long experience with ancient protein analysis, which is quite significant in this young research field, we have always strived to support our claims with full availability of raw data, use of negative controls and implementation of the highest best practice standards, for example by preparing samples in ancient DNA class clean laboratories to minimize cross-contaminations etc. For this reason, we are among those who led and inspired the definition of guidelines for best practice in ancient protein analysis (Hendy et al., Nature Ecology & Evolution 2018). Nevertheless, we would like to point out that, due to the fundamentally different nature of modern and ancient proteins, proteomics studies based on ancient proteins cannot be evaluated against the same quality standards used for ordinary proteomics studies of modern material. The reason is when analyzing ancient specimens, a number of unknown and uncontrollable random factors affect the samples investigated over an extended period of time. The scientists cannot control factors such as, among others: protein degradation, contamination, sample loss and lack of sample homogeneity, simply because they occurred before the samples became accessible for analysis. These limitations are intrinsic with the nature of ancient samples and not with negligence of

the scientists. In our opinion they should be taken into consideration even when datasets generated from ancient samples are compared with datasets generated from modern ones.

This study is indeed the first quantitative meta-proteomics study on ancient proteins and rejecting it based on expectations of quality criteria applied to modern proteomics studies would be unnecessarily and excessively severe, seriously limiting the progression and expansion of ancient protein research. On the contrary, we find it remarkable that despite all the uncontrolled factors randomly affecting ancient specimens, we can still see a statistical significant pattern with two ancient groups.

We hope the reviewer will find these considerations fair, reasonable and acceptable and that he/she will possibly reconsider our manuscript by adopting equally rigorous, but less overly demanding criteria.

Major concerns

1. Using unsupervised hierarchical clustering of protein LFQ intensity, the authors observed four clusters including modern plaque cluster, modern calculus cluster and two medieval clusters (G1 and G2). Is this clustering observation surprising? As indicated in the “Methods” section, the protein extraction and digestion methods are different for ancient and modern samples. Could that severely affect the sample clustering? The sum LFQ intensity of the samples varies >3 fold (as calculated from the database search output file: proteinGroups_workfile.txt). It seems the clustering also associated with the sum LFQ intensity variations. In addition, if I am correct, the raw file names indicate that the samples from different groups were analyzed on mass spectrometer as different batches. Whether the batch effects were evaluated? Could the clustering also be associated with the MS date?

We understand the reviewer’s concern and we will try to reason our analysis decisions in the following points:

The focus of this manuscript is the quantitative metaproteomic analysis of archaeological dental calculus. The comparison of the ancient material with modern dental calculus and dental plaque is purely indicative due to the admittedly different nature of the samples. As a consequence it is not surprising, although reassuring, that ancient dental calculus, modern dental calculus and modern dental plaque cluster separately. In our opinion, however, if not surprising, it is at least original, and not obvious at all, that the ancient dental calculus sample set splits in two distinct clusters, even after experiencing the effect of random degrading factors for more than five hundred years. Thus the main focus of the clustering is on the two subgroups within the Tjaerby samples. The modern samples were included, as outgroups, so that the reader could have an idea about the overlap of identified proteins between Tjaerby samples and modern samples.

- *The protein extraction protocol is identical for both the modern and medieval calculus samples. The protocol for protein extraction of the dental plaque samples obviously does not contain the initial demineralizing step since this step is unnecessary. Otherwise, only very minor details in the sample preparation, such as the concentrations in lysis buffer, are slightly different. Ultimately, the*

differences in the preparation of modern and ancient samples were either strictly necessary, e.g. demineralisation, due to the different nature of these two categories of samples, or very minor, e.g. lysis buffer concentration, with virtually no effect on the sample clustering. The summed LFQ intensity varies as you mention and it is likely affecting the clustering to some extent, but to best handle the issues of varying protein abundances between samples due to the fact that for archaeological samples one cannot control protein richness, degradation etc. we made use of the MaxLFQ algorithm for deriving protein intensities. The MaxLFQ algorithm was exactly developed to best determine and normalize protein abundance profiles by using the maximum possible information from MS-signals given that the presence of quantifiable peptides varies from sample to sample.

- *Also based on what mentioned above, we argue that batch effects should have a minimal, if any, effect on the sample clustering, considering random factors much more intense than this one extensively affected the ancient samples over centuries. Furthermore, it looks unlikely that batch effects can affect the metaproteomic composition of sample sets analysed in different days.*

All in all the clustering is meant as a visual aid to explain the differences we see in the two subgroups within the Tjaerby samples and the modern samples are only included to provide the reader with an overview of overlapping proteins.

2. The authors tried to link the oral metaproteome variations to oral health status in ancient individuals and made thorough discussions on the potential roles of G2-enriched commensals or G1-enriched potential pathogens. To further support the results, the calculus samples collected from healthy living individuals were also included for comparison. Why healthy individuals were used for the comparison, instead of periodontitis patients? It will be more convincing to also include a group of periodontitis patients since all the ancient samples displayed evidence of periodontitis (as indicated in Lines 66-68).

We agree that comparison to modern periodontitis patients would have been beneficial, but we did not have access to these types of samples. The reason is that today, fortunately, the average level of oral hygiene in Denmark makes the observation of periodontal disease cases as severe as those observed among medieval individuals extremely rare. Assembling a matched sample set of periodontitis patients would have not been easy and it would have required an effort beyond the scope of this manuscript. In this manuscript the focus is clearly pointed to the characterisation of ancient metaproteome. However, we agree that the comparison the reviewer suggests could be an interesting future investigation.

Again, we would like to repeat that the data from the modern samples is only used for comparisons and our main focus remains on the two subgroups within the Tjærby samples. We also believe that the literature about bacterial composition of periodontitis patients is already sufficiently rich to draw conclusions from, and consequently we did not find it appropriate to identify, analyze and include a cohort of periodontitis patients in this study.

3. Lines 293-294: “The modern plaque appears to be more variable than the modern calculus, possibly representing the dynamic nature of the ...”. It is not obvious to see the differences from the hierarchical clustering tree in Figure 2a. Supporting statistical analysis, such as the statistical comparison of the inter-individual distances, should be provided to make the above claims.

Thank you for this very important observation. After revision we realized that the statistics supporting our initial suggestion of variability was not strong enough, and after calculating beta diversity that is the difference in taxonomic abundance profiles from the different samples, we revised the manuscript accordingly. A supplementary figure of the beta diversity has now been included and the text corrected.

The paragraph has been changed from:

“Compared to the modern dental calculus samples, the bacterial profiles of G2 are the most similar, but both medieval groups show a much higher degree of inter-homogeneity than the modern individuals. **The modern plaque appears to be more variable than the modern calculus, possibly representing the dynamic nature of the uncalcified biofilm versus the mature and stabilized calculus.** The homogeneity of ancient calculus could reflect its lifelong deposition, or recovery may be biased towards the proteins that survive degradation.”

to:

“Compared to the modern dental calculus samples, the bacterial profiles of G2 are the most similar, but both medieval groups show a much higher degree of intra-homogeneity than the modern individuals (Supplementary figure 4). The homogeneity of ancient calculus could reflect its lifelong deposition, or recovery may be biased towards the proteins that survive degradation.”

furthermore a figure of the distribution of Bray-Curtis dissimilarity (below) has been added to supplementary

Supplementary Figure 4; Distribution of Bray-Curtis dissimilarity, with the value of 0 being similar and 1 being dissimilar. a) Distribution of Bray-Curtis dissimilarity within the groups. The Tjærby samples shows less diversity (more intra-homogeneity) than both the modern sample groups. b) Distribution of Bray-Curtis dissimilarity between modern calculus and the two subgroups of Tjærby samples showing Tjærby G2 to be more similar to modern calculus.

Minor comments:

1. Line 144: fig. 4d · Fig. 6

Thank you for spotting this inconsistency. The text has been edited from: “(Fig. 2a and fig. 4d)” to: “(Fig. 2a and fig. 6)”.

2. Figure 6: a set of 13 proteins with higher abundance in G2 compared to G1 were indicated, including those normal oral immune-related proteins specific to neutrophils. Are these proteins also higher abundance in the modern healthy samples? Why the modern sample groups were not included in the heat map?

These proteins are also of higher abundance in the modern samples, but most of the proteins are compared to the Tjærby samples. As already mentioned, the focus of the paper is not comparison to modern samples, but rather the use of quantitative data on archaeological samples, thus the modern samples were not included (and also the very different intensity values would make the differences in the Tjærby samples hard to visualize).

3. The TMT results seem to be distracting since the whole manuscript focus on the differences between the two subgroups of ancient calculus samples and their comparisons with samples collected from healthy living individuals.

Thank you for your comment. This was also discussed among the authors. After conducting the experiment we found the results quite intriguing although not as a complete data-set. We decided to include the TMT experiment to highlight a potentially strong future strategy for quantitative proteomics on archaeological samples. We agree that it is not in the direct scope of the paper and it is meant more as a future perspective.

We have therefore now moved the TMT figure (figure 8) to the supplementary figures and added the following to the introduction:

“A small pilot study of high pH fractionation in combination and TMT labelling, with the aim of identifying more proteins, was also included to show a potential future quantitative strategy.”

4. Figure 8: sample names should be labelled instead of the TMT channel names. More description in figure legend is needed. For example, how the fold changes were calculated (median was calculated by rows or columns)? Why is TMT131 obviously different with others?

Thank you for your comment. We have changed the labels on the figure and explained the differences in the figure text. All the clusterings in the manuscript are based on the same principle with log₂-transformation of the protein intensities and normalization by median subtraction (rows) to calculate fold-changes from the median, as also indicated in the figures.

Supplementary Figure 3; Hierarchical clustering of TMT labeled samples together with corresponding unlabeled sample. **Tjærby 6 shows a different profile from the other samples because this is the only sample belonging to G2 and it was one of the more protein rich samples.** The correlation between the labeled and unlabeled sample is good despite the missing values.

5. Details of the 7 healthy individuals are needed, such as the age and gender. Are they matched with the archaeological individuals?

Thank you for your observation. The information has been added to the manuscript.

The materials and method section (page ~20) has been changed from

“Supragingival plaque samples were collected from the oral surface of the mandibular incisors from seven healthy volunteers by use of a periodontal probe.”

to:

“Supragingival plaque samples were collected from the oral surface of the mandibular incisors from seven healthy volunteers (male=3, female=4, age ranging from 25 to 35 years) by use of a periodontal probe.”

The individuals were not matched with the archaeological individuals due to lack of appropriate volunteers.

Reviewer #3:

This study presents the results on dental calculus metaproteomes from 21 humans archaeologically recovered from a medieval (ca. 1100-1450 CE) cemetery at Tjaerby, Denmark, and constitutes the first attempt of quantitative metaproteomics of archaeological samples, following a seminal study by Warinner et al (2015). The authors state that metaproteomics has the potential to add new and quantitative levels of molecular information on the oral health status of individuals recovered from archaeological contexts. I was asked by the journal editors to comment on the bioarchaeological aspects of this study, and therefore the comments that follow have this domain as their sole focus.

The authors present novel results, of interest not only in assessing oral health in the specific population studied (medieval, ca. 1100-1450 CE cemetery at Tjaerby, Denmark), but also of clear methodological interest, both in terms of approaching oral health status of past individuals, but potentially also for investigating dietary proteins and milk consumption in particular, through the differentiation between bovine and caprine milk proteins found in calculus. As such, the results are of interest to the bioarchaeological research communities. While the work is convincing, some further information on the following aspects would improve the paper, its potential impact, and reproducibility:

We would like to thank the reviewer for the positive statements on our manuscript. We were very pleased to learn that the reviewer found our study to be of interest not only in assessing oral health in the specific population studied, but also of clear methodological interest, both in terms of approaching oral health status of past individuals, and as such, the results are of interest to the bioarchaeological research communities.

(1) Why were only male individuals included in the study? While there may be a good reason for this, it is currently not stated explicitly in the paper. This needs to be discussed. What is also not clear is whether the authors considered (or will consider) making similar analyses for females, and whether there are calculus deposits which would allow subadults to be analysed.

Older males were chosen because we wanted to identify the most advanced and obvious cases of periodontal disease. Initially, only the oldest males (i.e. over 45) were to be used, preservation limited this to 12 individuals, so younger males showing periodontitis were also included. We were looking for cases where 'periodontal troughs' had formed and advanced periodontal disease could be surmised. These selection criteria have been expanded on in the introduction. There is no plan to include female or subadults skeletons at this time, as this was out of the original scope of the project. As for non-adults, periodontitis is quite uncommon in children and teenagers, and although population specific, dental calculus is usually insufficient for sampling. Having examined a number of non-adults from Tjærby, the bioarchaeologist on the project (L.T.L.) can

confirm this to be the case. No female individuals were included in the study to avoid the introduction of another variable in the quantitative analysis of the proteomic data. Considering how innovative the application of quantitative proteomic analysis to archaeological samples is, we initially chose to proceed with the most favourable sample set. Based on the results obtained, we can confidently claim our approach could be reliably used in the future on sample sets from individuals of both sexes.

Added to introduction:

“In order to look at the diversity in microbiome species, as well as disease biomarkers from dental calculus, we were interested in individuals most likely to have osseous changes and destruction of the alveolar bone related to periodontitis. Older males were selected based on the understanding that males often have a more aggressive inflammatory immune response compared to females¹⁵⁻¹⁷. Males over 45 with a maxilla, mandible, teeth, evidence of periodontal disease, and sufficient dental calculus for sampling, were the initial criteria for inclusion. However, this yielded only 12 samples. By adding younger males, it was possible to collect 22 dental calculus samples from 21 individuals.”

(2) Why does the study include only middle/older adults, save for one younger adult individual? Again, there may be a good reason for this (e.g. insufficient calculus deposits?), but this needs to be discussed explicitly in the paper.

Thank you for your comment. We hope that the paragraph added in the introduction, and mentioned in our reply to the previous comment, answers your question.

“In order to look at the diversity in microbiome species, as well as disease biomarkers from dental calculus, we were interested in individuals most likely to have osseous changes and destruction of the alveolar bone related to periodontitis. Older males were selected based on the understanding that males often have a more aggressive inflammatory immune response compared to females¹⁵⁻¹⁷. Males over 45 with a maxilla, mandible, teeth, evidence of periodontal disease, and sufficient dental calculus for sampling, were the initial criteria for inclusion. However, this yielded only 12 samples. By adding younger males, it was possible to collect 22 dental calculus samples from 21 individuals.”

(3) As to reproducibility of this kind of work, it would be very useful to have more information on the calculus deposits. The bioarchaeological reader would benefit from a fuller description (e.g. in the supplementary data) of the calculus deposits in each dentition, their location and character (e.g. supra- or subgingival), as well as extent. At minimum, at least one illustration of a calculus deposit could be included, e.g. within the sampling protocol figure, or within the supplementary materials. Further, the choices as to where an individual’s calculus deposits were (preferentially) sampled from is not clear.

A full description of dental calculus samples (i.e. tooth, location, weight) is included in Supplementary Table 2; however, inferring whether calculus is supra- or subgingival in

archaeological remains is extremely challenging, especially where gingival recession is advanced. Subsequently, locations of sampling are listed as crown, root, or mixed. Sampling strategy details have been added to the introduction and in supplementary information 1 we have added:

“Samples were taken from the largest deposits available, but sometimes samples had to be aggregated, as a single tooth did not have enough material available (see Supplementary Table 2).”

plus a photo of typical calculus for sampling (1C).

Supplementary Table 2; Dental calculus sample notes.

Sample/ Accession number for each individual	Sample Location	Teeth Sampled	Sample Weight
#1 A795	Crown / root	Buccal UR8 and labial root LR6	18.2 mg
#2 A1408	Crown	Distal and lingual ‘verrucous’ calculus extending onto the roots from the LL8	145.4 mg
#3 A1416	Crown / root	Distal UL8, buccal roots UR6, buccal roots UL7	15.5 mg
#4 A1637	Crown	Lingual and buccal LL7 and LL1	42.5 mg
#5 A1671	Crown	Lingual LL1	19.2 mg
#6 A1673	Crown	Lingual LR1	31.0 mg
#7 A1899	Crown	Lingual LL5, LL8, and LR2	37.7 mg
#8 BLANK			
#9 A1294	Crown / root	Distal supragingival UL8 and buccal roots UR6	23.0 mg
#10 A1408	Crown	‘Normal’ sample, labial UR3.	25.9 mg
#11 A1442	Crown	Lingual and buccal LL7	34.2 mg
#12 A1623	Crown / root	Buccal CEJ LR2, LR3, LR4, and LR5	15.6 mg
#13 A1635	Crown	Buccal and lingual UR7 and Lingual UR6	32.9 mg

#14 A1764	Crown	Labial UR2 and UR3	39.0 mg
#15 A1893	Crown	Buccal UL7 and UR4	25.9 mg
#16 A1898	Crown	Buccal and lingual UL6, UR1, and UR2	30.1 mg
#17 BLANK			
#18 A983*	Root	UR7	127.0 mg
#19 A970	Crown	Labial LR4, LR6, and distal LR8	26.1 mg
#20 A1235	Crown	Distal and mesial UR8	23.2 mg
#21 A1453	Crown / root	LR3 (lingual root, M/D/B crown) and LL2 lingual root	31.2 mg
#22 A1866	Crown	Buccal and lingual LR4	45.9 mg
#23 A1664	Crown	Buccal LR3 and LR1	35.8 mg
#24 A1968	Root	Buccal root LL6	34.0 mg
#25 BLANK			

Supplementary 1C) Sampling location of dental calculus from Tjærby #22 (A1866).

(4) Photograph/s of a representative dentition and alveoli showing periodontal disease within this population would be useful. If the maximum number of illustrations has already been reached, such an image could be included within the sampling protocol figure as one of the sub-figures.

We agree and have added the following photos to supplementary information 1.

Supplementary 1D) The photograph of #6 (A1673) showing periodontal troughs along the buccal edge of the left molars (note also glue in the socket of LL5).

Supplementary 1E) A large carious lesion that has destroyed the crown of the upper right first molar with associated abscess formation in individual #7 (A1899)

Supplementary 1F) Large abscess corresponding to the upper right first molar of #11 (A1442).

(5) The importance of the medieval site is not elaborated sufficiently for the reader to get a sense of why this medieval parish site has any importance beyond a mere case study. Why should we learn more about the oral health of a medieval village population in Denmark? What are the larger (archaeological/human health) questions here? What is the impact of the results derived from this particular site – or could any assemblage with individuals with sufficient amount of calculus have been chosen? The authors state that '[t]he site is of great interest because it has been fully excavated, and the skeletons reflect a typical, medieval village parish population', but do not qualify this statement by giving information on how many (contemporary) sites in Denmark have been completely excavated, and whether any other medieval village parish populations are available for study. An explicit statement on why the study of oral health within a typical, medieval village parish population is of importance would be welcome. Given that only a relatively small number of skeletons (n=21, all male) from the overall population was analysed it may be of lesser importance that the site was fully excavated. The paper needs to elaborate more fully on the importance/uniqueness of the site and/or the skeletal population/assemblage, and the wider questions that can be explored through them, or state that the analyses could have been conducted on any other (contemporary) skeletal assemblage to illustrate the potential of the method.

Thanks for raising this very relevant point. We focused our attention on the Tjærby site because of the high quality standards adopted for its excavation and for the conservation of the human remains collected from the site. The accurate documentation

associated with each human remain investigated avoided methodological errors easily incurred when the archaeological context is not so well documented. As mentioned in the replies to comments from reviewer #2, there are already several random factors that can affect the reconstruction of quantitative proteomics patterns, and we strived to avoid an extra major one. Apart from that, the Tjærby site can be considered an ordinary example of medieval human occupation in Denmark. For this reason, the approach we used for the analysis of the human remains from this site could be easily extended to other Scandinavian and to other European sites from the same time range or more ancient assemblages. This study should be considered a methodological pilot in preparation of a larger survey of the entire Scandinavian area from the Neolithic to the middle ages. The extraction and mass spectrometric analysis of hundreds of samples is currently in progress. The larger archaeological question here is the reconstruction of the relevance of oral disease in Northern Europe, over a time range of a few millennia.

Amended information about the site in the introduction:

“The site represents an ordinary Danish medieval village, and thus the individuals should show a relative degree of uniformity in terms of lifeways and social status, allowing the recovery of proteomes that are fairly comparable.

This site was chosen because the material is well-curated and easily accessible, but also of interest because it is the only Danish cemetery from the medieval period that has been fully excavated. Tjærby in its ‘mundaneness’ - gives insight into the Danish medieval oral microbiomes of a relatively large number of ‘average’ individuals. By studying individual oral microbiomes, future comparison with other assemblages, either contemporary or more ancient, should be more nuanced”

Additional details have been added to the supplementary information 1A.

“The cemetery of Tjærby was chosen as the material is well-curated, and because it represents a complete medieval assemblage. This allows for larger numbers of remains (in this case older males), to be analysed from an intra-site perspective. Being medieval, preservation of biomolecules should theoretically be better than earlier periods, but the material also predates major changes to diet and lifestyle that occur in the post medieval period. Social stratification in a rural parish cemetery should be limited. Therefore, through this relative homogeneity, we can assess as closely as possible to an ‘average medieval person’s’ microbiome.”

The approach taken by the paper is potentially of great interest to the bioarchaeological community, and may influence how we approach past oral health in the future.

REVIEWERS' COMMENTS:

Reviewer #1 (Remarks to the Author):

I appreciate the authors' thoughtful responses to my original reviews to this manuscript. I think the changes cover most of my concerns. I think the diagrams in Table 1 are much clearer with the thicker line width and the addition of representative figures in the supplement really clarifies the text. Based on the changes, I have a few additional things I would like to see addressed.

Page 20, Line 363-367: Can you calculate what the average age of individuals used were or are the individual ages estimates based on osteological correlates?

Page 21, Line 401: Please list what the average age and standard deviation of the living individuals is. Please justify why female samples were taken when only male archeological samples were utilized. With respect to reviewer 2's comment about age matching, please explain why similar aged individuals were not used for modern controls. Also, can you detect differences in your data as a result of age, or are the differences strictly related to oral microbome?

Figure 3: Please color Desulfomicrobium to match Figure 2.

Supplementary figure 1D: Please add arrows to clarify where the periodontal troughs are.

Supplementary figure 1E-F: Please add arrows to show where the abscesses and/or carious lesions are. This will make it easier to understand for non-experts.

Reviewer #3 (Remarks to the Author):

Thank you for your response. I consider all concerns to have been addressed satisfactorily.

NCOMMS-18-03009A

“Quantitative metaproteomics of medieval dental calculus reveals individual oral health status” by *Jersie-Christensen et al.*

Our responses to each of the reviewer points are indicated in *italicized blue text*.

New references added to the manuscript are indicated in quotation marks below responses.

REVIEWERS' COMMENTS:

Reviewer #1 (Remarks to the Author):

I appreciate the authors' thoughtful responses to my original reviews to this manuscript. I think the changes cover most of my concerns. I think the diagrams in Table 1 are much clearer with the thicker line width and the addition of representative figures in the supplement really clarifies the text. Based on the changes, I have a few additional things I would like to see addressed.

Page 20, Line 363-367: Can you calculate what the average age of individuals used were or are the individual ages estimates based on osteological correlates?

Individual ages are based on osteological techniques examining morphological changes to bones (namely the pelvis) (see Buikstra and Ubelaker, 1994), as well as factors like dental wear. Although some methodologies propose much smaller age ranges (i.e. with 5 year intervals) (i.e. Lovejoy et. al), the lead bioarchaeologist in the study (LTL) believes these do not adequately take into consideration individual variation and lifestyle. Osseous changes to the areas such as the pubic symphysis and the auricular surface, do correlate to advancing age, but it is usually impossible to separate a 'long life' from a 'hard life.' Other factors like dental wear become more advanced with age, but are highly dependent on the nature of the diet. Therefore, more broad age categories should be used and grouped accordingly (see Powers, 2012), because these take into consideration lifestyle and population differences. The +45 category is particularly 'inaccurate,' and it would be inappropriate, and impossible, to calculate an average age for unknown archaeological individuals.

Buikstra, J.E. and Ubelaker D.H. Standards for Data Collection from Human Skeletal Remains. Arkansas Archaeological Survey Research Series No. 44, Fayetteville.

Lovejoy, C. O., Meindl, R. S., Pryzbeck, T. R. and Mensforth, R. P. (1985), Chronological metamorphosis of the auricular surface of the ilium: A new method for the determination of adult skeletal age at death. Am. J. Phys. Anthropol., 68: 15-28. doi:10.1002/ajpa.1330680103

Powers, N. Human osteology method statement - Museum of London. (Museum of London, 2012).

Page 21, Line 401: Please list what the average age and standard deviation of the living individuals is. Please justify why female samples were taken when only male archeological samples were utilized. With respect to reviewer 2's previous comment about age matching, please explain why similar aged individuals were not used for modern controls. Also, can you detect differences in your data as a result of age, or are the differences strictly related to oral microbome?

We thank the reviewer for this suggestion and have now added information about the average age in the manuscript text in the Methods section that now reads:

“Supragingival plaque samples were collected from the oral surface of the mandibular incisors from seven healthy volunteers (male=3, female=4, age ranging from 25 to 35 years with an average of 29 years) by use of a periodontal probe.”

Note that the modern subjects were not matched with the archaeological individuals due to lack of appropriate volunteers. We tried to correlate all the archaeological samples to as many parameters as possible; age, AMTL, carries, Bradford score and abscesses but did not find a clear correlation to any of these. Thus, the grouping of the individuals is solely based on the proteomics data and this is indeed why find it very interesting, because it could add another level of information not only based on e.g. osteological observations.

Figure 3: Please color *Desulfomicrobium* to match Figure 2.

Done. See updated figure 3 here:

Figure 3

Supplementary figure 1D: Please add arrows to clarify where the periodontal troughs are.

Done. See updated figure here:

Supplementary figure 1E-F: Please add arrows to show where the abscesses and/or carious lesions are. This will make it easier to understand for non-experts.

Done. See updated figures here:

Reviewer #3 (Remarks to the Author):

Thank you for your response. I consider all concerns to have been addressed satisfactorily.

We thank the reviewer for acknowledging our work with the revised manuscript.